# Optimal ANN-SNN Conversion for High-accuracy and Ultra-low-latency Spiking Neural Networks

**Tong Bu**[1], **Wei Fang**[1], **Jianhao Ding**[1], **PengLin Dai**[2], **Zhaofei Yu**[1 *], **Tiejun Huang**[1]
[1] Peking University, [2] Southwest Jiaotong University
[*] Corresponding author: yuzf12@pku.edu.cn

### Abstract

Spiking Neural Networks (SNNs) have gained great attraction due to their distinctive properties of low power consumption and fast inference on neuromorphic hardware. As the most effective method to get deep SNNs, ANN-SNN conversion has achieved comparable performance as ANNs on large-scale datasets. Despite this, it requires long time-steps to match the firing rates of SNNs to the activation of ANNs. As a result, the converted SNN suffers severe performance degradation problems with short time-steps, which hamper the practical application of SNNs. In this paper, we theoretically analyze ANN-SNN conversion error and derive the estimated activation function of SNNs. Then we propose the quantization clip-floor-shift activation function to replace the ReLU activation function in source ANNs, which can better approximate the activation function of SNNs. We prove that the expected conversion error between SNNs and ANNs is zero, enabling us to achieve high-accuracy and ultra-low-latency SNNs. We evaluate our method on CIFAR-10/100 and ImageNet datasets, and show that it outperforms the state-of-the-art ANN-SNN and directly trained SNNs in both accuracy and time-steps. To the best of our knowledge, this is the first time to explore high-performance ANN-SNN conversion with ultra-low latency (4 time-steps). Code is available at https://github.com/putshua/SNN_conversion_QCFS

## 1 Introduction

Spiking neural networks (SNNs) are biologically plausible neural networks based on the dynamic characteristic of biological neurons (McCulloch & Pitts, 1943; Izhikevich, 2003). As the third generation of artificial neural networks (Maass, 1997), SNNs have attracted great attention due to their distinctive properties over deep analog neural networks (ANNs) (Roy et al., 2019). Each neuron transmits discrete spikes to convey information when exceeding a threshold. For most SNNs, the spiking neurons will accumulate the current of the last layer as the output within $T$ inference time steps. The binarized activation has rendered dedicated hardware of neuromorphic computing (Pei et al., 2019; DeBole et al., 2019; Davies et al., 2018). This kind of hardware has excellent advantages in temporal resolution and energy budget. Existing work has shown the potential of tremendous energy saving with considerably fast inference (Stöckl & Maass, 2021).

In addition to efficiency advantages, the learning algorithm of SNNs has been improved by leaps and bounds in recent years. The performance of SNNs trained by backpropagation through time and ANN-SNN conversion techniques has gradually been comparable to ANNs on large-scale datasets (Fang et al., 2021; Rueckauer et al., 2017). Both techniques benefit from the setting of SNN inference time. Setting longer time-steps in backpropagation can make the gradient of surrogate functions more reliable (Wu et al., 2018; Neftci et al., 2019; Zenke & Vogels, 2021). However, the price is enormous resource consumption during training. Existing platforms such as TensorFlow and PyTorch based on CUDA have limited optimization for SNN training. In contrast, ANN-SNN conversion usually depends on a longer inference time to get comparable accuracy as the original ANN (Sengupta et al., 2019) because it is based on the equivalence of ReLU activation and integrate-and-fire model's firing rate (Cao et al., 2015). Although longer inference time can further reduce the conversion error, it also hampers the practical application of SNNs on neuromorphic chips.

The dilemma of ANN-SNN conversion is that there exists a remaining potential in the conversion theory, which is hard to be eliminated in a few time steps (Rueckauer et al., 2016). Although many methods have been proposed to improve the conversion accuracy, such as weight normalization (Diehl et al., 2015), threshold rescaling (Sengupta et al., 2019), soft-reset (Han & Roy, 2020) and threshold shift (Deng & Gu, 2020), tens to hundreds of time-steps in the baseline works are still unbearable. To obtain high-performance SNNs with ultra-low latency (e.g., 4 time-steps), we list the critical errors in ANN-SNN conversion and provide solutions for each error. Our main contributions are summarized as follows:

- We go deeper into the errors in the ANN-SNN conversion and ascribe them to clipping error, quantization error, and unevenness error. We find that unevenness error, which is caused by the changes in the timing of arrival spikes and has been neglected in previous works, can induce more spikes or fewer spikes as expected.

- We propose the quantization clip-floor-shift activation function to replace the ReLU activation function in source ANNs, which better approximates the activation function of SNNs. We prove that the expected conversion error between SNNs and ANNs is zero, indicating that we can achieve high-performance converted SNN at ultra-low time-steps.

- We evaluate our method on CIFAR-10, CIFAR-100, and ImageNet datasets. Compared with both ANN-SNN conversion and backpropagation training methods, the proposed method exceeds state-of-the-art accuracy with fewer time-steps. For example, we reach top-1 accuracy 91.18% on CIFAR-10 with unprecedented 2 time-steps.

## 2 PRELIMINARIES

In this section, we first briefly review the neuron models for SNNs and ANNs. Then we introduce the basic framework for ANN-SNN conversion.

**Neuron model for ANNs.** For ANNs, the computations of analog neurons can be simplified as the combination of a linear transformation and a non-linear mapping:

$$\boldsymbol{a}^l = h(\boldsymbol{W}^l \boldsymbol{a}^{l-1}), \quad l = 1, 2, ..., M \tag{1}$$

where the vector $\boldsymbol{a}^l$ denotes the output of all neurons in $l$-th layer, $\boldsymbol{W}^l$ denotes the weight matrix between layer $l$ and layer $l - 1$, and $h(\cdot)$ is the ReLU activation function.

**Neuron model for SNNs.** Similar to the previous works (Cao et al., 2015; Diehl et al., 2015; Han et al., 2020), we consider the Integrate-and-Fire (IF) model for SNNs. If the IF neurons in $l$-th layer receive the input $\boldsymbol{x}^{l-1}(t)$ from last layer, the temporal potential of the IF neurons can be defined as:

$$\boldsymbol{m}^l(t) = \boldsymbol{v}^l(t-1) + \boldsymbol{W}^l \boldsymbol{x}^{l-1}(t), \tag{2}$$

where $\boldsymbol{m}^l(t)$ and $\boldsymbol{v}^l(t)$ represent the membrane potential before and after the trigger of a spike at time-step $t$. $\boldsymbol{W}^l$ denote the weight in $l$-th layer. As soon as any element $m_i^l(t)$ of $\boldsymbol{m}^l(t)$ exceeds the firing threshold $\theta^l$, the neuron will elicit a spike and update the membrane potential $v_i^l(t)$. To avoid information loss, we use the "reset-by-subtraction" mechanism (Rueckauer et al., 2017; Han et al., 2020) instead of the "reset-to-zero" mechanism, which means the membrane potential $v_i^l(t)$ is subtracted by the threshold value $\theta^l$ if the neuron fires. Based on the threshold-triggered firing mechanism and the "reset-by-subtraction" of the membrane potential after firing discussed above, we can write the uplate rule of membrane potential as:

$$\boldsymbol{s}^l(t) = H(\boldsymbol{m}^l(t) - \boldsymbol{\theta}^l), \tag{3}$$

$$\boldsymbol{v}^l(t) = \boldsymbol{m}^l(t) - \boldsymbol{s}^l(t)\theta^l. \tag{4}$$

Here $\boldsymbol{s}^l(t)$ refers to the output spikes of all neurons in layer $l$ at time $t$, the element of which equals 1 if there is a spike and 0 otherwise. $H(\cdot)$ is the Heaviside step function. $\boldsymbol{\theta}^l$ is the vector of the firing threshold $\theta^l$. Similar to Deng & Gu (2020), we suppose that the postsynaptic neuron in $l$-th layer receives unweighted postsynaptic potential $\theta^l$ if the presynaptic neuron in $l - 1$-th layer fires a spike, that is:

$$\boldsymbol{x}^l(t) = \boldsymbol{s}^l(t)\theta^l. \tag{5}$$

Table 1: Summary of notations in this paper

| Symbol | Definition | Symbol | Definition |
|--------|-----------|--------|-----------|
| $l$ | Layer index | $\boldsymbol{x}^l(t)$ | Unweighted PSP[1] |
| $i$ | Neuron index | $\boldsymbol{s}^l(t)$ | Output spikes |
| $\boldsymbol{W}^l$ | Weight | $\boldsymbol{\phi}^l(T)$ | Average unweigthed PSP before time $T$ |
| $\boldsymbol{a}^l$ | ANN activation values | $\boldsymbol{z}^l$ | Weighted input from $l-1$ layer |
| $t$ | Time-steps | $h(\cdot)$ | ReLU function |
| $T$ | Total time-step | $H(\cdot)$ | Heaviside step function |
| $\boldsymbol{\theta}^l$ | Threshold | $L$ | Quantization step for ANN |
| $\lambda^l$ | Trainable threshold in ANN | $\boldsymbol{Err}^l$ | Conversion Error |
| $\boldsymbol{m}^l(t)$ | Potential before firing | $\widetilde{\boldsymbol{Err}}^l$ | Estimated conversion Error |
| $\boldsymbol{v}^l(t)$ | Potential after firing | $\varphi$ | Shift of quantization clip-floor function |

[1] Postsynaptic potential

**ANN-SNN conversion.** The key idea of ANN-SNN conversion is to map the activation value of an analog neuron in ANN to the firing rate (or average postsynaptic potential) of a spiking neuron in SNN. Specifically, we can get the potential update equation by combining Equation 2 – Equation 4:

$$\boldsymbol{v}^l(t) - \boldsymbol{v}^l(t-1) = \boldsymbol{W}^l \boldsymbol{x}^{l-1}(t) - \boldsymbol{s}^l(t)\theta^l. \tag{6}$$

Equation 6 describes the basic function of spiking neurons used in ANN-SNN conversion. By summing Equation 6 from time 1 to $T$ and dividing $T$ on both sides, we have:

$$\frac{\boldsymbol{v}^l(T) - \boldsymbol{v}^l(0)}{T} = \frac{\boldsymbol{W}^l \sum_{i=1}^{T} \boldsymbol{x}^{l-1}(i)}{T} - \frac{\sum_{i=1}^{T} \boldsymbol{s}^l(i)\theta^l}{T}. \tag{7}$$

If we use $\boldsymbol{\phi}^{l-1}(T) = \frac{\sum_{i=1}^{T} \boldsymbol{x}^{l-1}(i)}{T}$ to denote the average postsynaptic potential during the period from 0 to $T$ and substitute Equation 5 into Equation 7, then we get:

$$\boldsymbol{\phi}^l(T) = \boldsymbol{W}^l \boldsymbol{\phi}^{l-1}(T) - \frac{\boldsymbol{v}^l(T) - \boldsymbol{v}^l(0)}{T}. \tag{8}$$

Equation 8 describes the relationship of the average postsynaptic potential of neurons in adjacent layers. Note that $\boldsymbol{\phi}^l(T) \geqslant 0$. If we set the initial potential $\boldsymbol{v}^l(0)$ to zero and neglect the remaining term $\frac{\boldsymbol{v}^l(T)}{T}$ when the simulation time-steps $T$ is long enough, the converted SNN has nearly the same activation function as source ANN (Equation 1). However, high $T$ would cause long inference latency that hampers the practical application of SNNs. Therefore, this paper aims to implement high-performance ANN-SNN conversion with extremely low latency.

## 3 CONVERSION ERROR ANALYSIS

In this section, we will analyze the conversion error between the source ANN and the converted SNN in each layer in detail. In the following, we assume that both ANN and SNN receive the same input from the layer $l-1$, that is, $\boldsymbol{a}^{l-1} = \boldsymbol{\phi}^{l-1}(T)$, and then analyze the error in layer $l$. For simplicity, we use $\boldsymbol{z}^l = \boldsymbol{W}^l \boldsymbol{\phi}^{l-1}(T) = \boldsymbol{W}^l \boldsymbol{a}^{l-1}$ to substitute the weighted input from layer $l-1$ for both ANN and SNN. The absolute conversion error is exactly the outputs from converted SNN subtract the outputs from ANN:

$$\boldsymbol{Err}^l = \boldsymbol{\phi}^l(T) - \boldsymbol{a}^l = \boldsymbol{z}^l - \frac{\boldsymbol{v}^l(T) - \boldsymbol{v}^l(0)}{T} - h(\boldsymbol{z}^l), \tag{9}$$

where $h(\boldsymbol{z}^l) = \text{ReLU}(\boldsymbol{z}^l)$. It can be found from Equation 9 that the conversion error is nonzero if $\boldsymbol{v}^l(T) - \boldsymbol{v}^l(0) \neq 0$ and $\boldsymbol{z}^l > 0$. In fact, the conversion error is caused by three factors.

**Clipping error.** The output $\boldsymbol{\phi}^l(T)$ of SNNs is in the range of $[0, \theta^l]$ as $\boldsymbol{\phi}^l(T) = \frac{\sum_{i=1}^{T} \boldsymbol{x}^l(i)}{T} = \frac{\sum_{i=1}^{T} \boldsymbol{s}^l(i)}{T}\theta^l$ (see Equation 5). However, the output $\boldsymbol{a}^l$ of ANNs is in a much lager range of $[0, a^l_{max}]$, where $a^l_{max}$ denotes the maximum value of $\boldsymbol{a}^l$. As illustrated in Figure 1a, $\boldsymbol{a}^l$ can be mapped to $\boldsymbol{\phi}^l(T)$ by the following equation:

$$\boldsymbol{\phi}^l(T) = \text{clip}\left(\frac{\theta^l}{T} \left\lfloor \frac{\boldsymbol{a}^l T}{\lambda^l} \right\rfloor, 0, \theta^l\right). \tag{10}$$

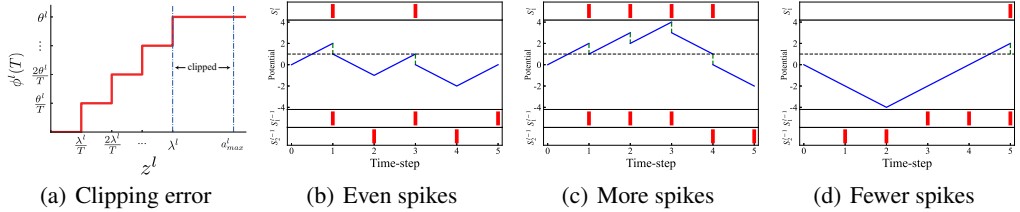

(a) Clipping error      (b) Even spikes      (c) More spikes      (d) Fewer spikes

Figure 1: Conversion error between source ANN and converted SNN. $s_1^{l-1}$ and $s_2^{l-1}$ denote the output spikes of two neurons in layer $l-1$, and $s_1^l$ denotes the output spikes of a neuron in layer $l$.

Here the clip function sets the upper bound $\theta^l$ and the lower bound $0$. $\lfloor \cdot \rfloor$ denotes the floor function. $\lambda^l$ represents the actual maximum value of output $\boldsymbol{a}^l$ mapped to the maximum value $\theta^l$ of $\boldsymbol{\phi}^l(T)$. Considering that nearly 99.9% activations of $\boldsymbol{a}^l$ in ANN are in the range of $[0, \frac{a_{max}^l}{3}]$, Rueckauer et al. (2016) suggested to choose $\lambda^l$ according to 99.9% activations. The activations between $\lambda^l$ and $a_{max}^l$ in ANN are mapped to the same value $\theta^l$ in SNN, which will cause conversion error called clipping error.

**Quantization error (flooring error).** The output spikes $\boldsymbol{s}^l(t)$ are discrete events, thus $\boldsymbol{\phi}^l(T)$ are discrete with quantization resolution $\frac{\theta^l}{T}$ (see Equation 10). When mapping $\boldsymbol{a}^l$ to $\boldsymbol{\phi}^l(T)$, there exists unavoidable quantization error. For example, as illustrated in Figure 1a, the activations of ANN in the range of $[\frac{\lambda^l}{T}, \frac{2\lambda^l}{T})$ are mapped to the same value $\frac{\theta^l}{T}$ of SNN.

**Unevenness error.** Unevenness error is caused by the unevenness of input spikes. If the timing of arrival spikes changes, the output firing rates may change, which causes conversion error. There are two situations: more spikes as expected or fewer spikes as expected. To see this, in source ANN, we suppose that two analog neurons in layer $l-1$ are connected to an analog neuron in layer $l$ with weights 2 and -2, and the output vector $\boldsymbol{a}^{l-1}$ of neurons in layer $l-1$ is $[0.6, 0.4]$. Besides, in converted SNN, we suppose that the two spiking neurons in layer $l-1$ fire 3 spikes and 2 spikes in 5 time-steps (T=5), respectively, and the threshold $\theta^{l-1} = 1$. Thus, $\phi^{l-1}(T) = \frac{\sum_{i=1}^{T} \boldsymbol{s}^{l-1}(i)}{T}\theta^{l-1} = [0.6, 0.4]$. Even though $\phi^{l-1}(T) = \boldsymbol{a}^{l-1}$ and the weights are same for the ANN and SNN, $\phi^l(T)$ can be different from $\boldsymbol{a}^l$ if the timing of arrival spikes changes. According to Equation 1, the ANN output $\boldsymbol{a}^l = \boldsymbol{W}^l \boldsymbol{a}^{l-1} = [2, -2][0.6, 0.4]^T = 0.4$. As for SNN, supposing that the threshold $\theta^l = 1$, there are three possible output firing rates, which are illustrated in Figure 1 (b)-(d). If the two presynaptic neurons fires at $t = 1, 3, 5$ and $t = 2, 4$ (red bars) respectively with weights 2 and -2, the postsynaptic neuron will fire two spikes at $t = 1, 3$ (red bars), and $\phi^l(T) = \frac{\sum_{i=1}^{T} \boldsymbol{s}^l(i)}{T}\theta^l = 0.4 = \boldsymbol{a}^l$. However, if the presynaptic neurons fires at $t = 1, 2, 3$ and $t = 4, 5$, respectively, the postsynaptic neuron will fire four spikes at $t = 1, 2, 3, 4$, and $\phi^l(T) = 0.8 > \boldsymbol{a}^l$. If the presynaptic neurons fires at $t = 3, 4, 5$ and $t = 1, 2$, respectively, the postsynaptic neuron will fire only one spikes at $t = 5$, and $\phi^l(T) = 0.2 < \boldsymbol{a}^l$.

Note that the clipping error and quantization error have been proposed in Li et al. (2021). There exist interdependence between the above three kinds of errors. Specifically, the unevenness error will degenerate to the quantization error if $\boldsymbol{v}^l(T)$ is in the range of $[0, \theta^l]$. Assuming that the potential $\boldsymbol{v}^l(T)$ falls into $[0, \theta^l]$ will enable us to estimate the activation function of SNNs ignoring the effect of unevenness error. Therefore, an estimation of the output value $\phi^l(T)$ in a converted SNN can be formulated with the combination of clip function and floor function, that is:

$$\boldsymbol{\phi}^l(T) \approx \theta^l \ \mathrm{clip}\left(\frac{1}{T}\left\lfloor \frac{\boldsymbol{z}^l T + \boldsymbol{v}^l(0)}{\theta^l}\right\rfloor, 0, 1\right). \tag{11}$$

The detailed derivation is in the Appendix. With the help of this estimation for the SNN output, the estimated conversion error $\widetilde{\boldsymbol{Err}}^l$ can be derived from Equation 9:

$$\widetilde{\boldsymbol{Err}}^l = \theta^l \ \mathrm{clip}\left(\frac{1}{T}\left\lfloor \frac{\boldsymbol{z}^l T + \boldsymbol{v}^l(0)}{\theta^l}\right\rfloor, 0, 1\right) - h(\boldsymbol{z}^l) \approx \boldsymbol{Err}^l. \tag{12}$$

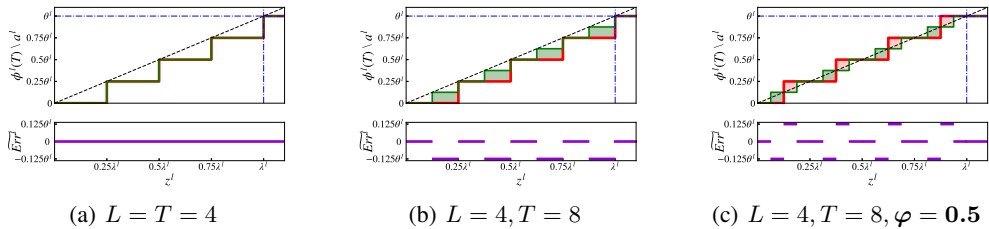

(a) $L = T = 4$       (b) $L = 4, T = 8$      (c) $L = 4, T = 8, \boldsymbol{\varphi} = \mathbf{0.5}$

Figure 2: Comparison of SNN output $\boldsymbol{\phi}^l(T)$ and ANN output $\boldsymbol{a}^l$ with same input $\boldsymbol{z}^l$

## 4 OPTIMAL ANN-SNN CONVERSION

### 4.1 QUANTIZATION CLIP-FLOOR ACTIVATION FUNCTION

According to the conversion error of Equation 12, it is natural to think that if the commonly used ReLU activation function $h(\boldsymbol{z}^l)$ is substituted by a clip-floor function with a given quantization steps $L$ (similar to Equation 11), the conversion error at time-steps $T = L$ will be eliminated. Thus the performance degradation problem at low latency will be solved. As shown in Equation 13, we proposed the quantization clip-floor activation function to train ANNs.

$$\boldsymbol{a}^l = \bar{h}(\boldsymbol{z}^l) = \lambda^l \, \text{clip} \left( \frac{1}{L} \left\lfloor \frac{\boldsymbol{z}^l L}{\lambda^l} \right\rfloor, 0, 1 \right), \tag{13}$$

where the hyperparameter $L$ denotes quantization steps of ANNs, the trainable $\lambda^l$ decides the maximum value of $\boldsymbol{a}^l$ in ANNs mapped to the maximum of $\boldsymbol{\phi}^l(T)$ in SNNs. Note that $\boldsymbol{z}^l = \boldsymbol{W}^l \boldsymbol{\phi}^{l-1}(T) = \boldsymbol{W}^l \boldsymbol{a}^{l-1}$. With this new activation function, we can prove that the estimated conversion error between SNNs and ANNs is zero, and we have the following Theorem.

**Theorem 1.** *An ANN with activation function (13) is converted to an SNN with the same weights. If $T = L$, $\theta^l = \lambda^l$, and $\boldsymbol{v}^l(0) = \mathbf{0}$, then:*

$$\widetilde{\boldsymbol{Err}}^l = \boldsymbol{\phi}^l(T) - \boldsymbol{a}^l = \mathbf{0}. \tag{14}$$

*Proof.* According to Equation 12, and the conditions $T = L$, $\theta^l = \lambda^l$, $\boldsymbol{v}^l(0) = \mathbf{0}$, we have $\widetilde{\boldsymbol{Err}}^l = \boldsymbol{\phi}^l(T) - \boldsymbol{a}^l = \theta^l \, \text{clip} \left( \frac{1}{T} \left\lfloor \frac{\boldsymbol{z}^l T + \boldsymbol{v}^l(0)}{\theta^l} \right\rfloor, 0, 1 \right) - \lambda^l \, \text{clip} \left( \frac{1}{L} \left\lfloor \frac{\boldsymbol{z}^l L}{\lambda^l} \right\rfloor, 0, 1 \right) = 0.$ $\square$

Theorem 1 implies that if the time-steps $T$ of the converted SNN is the same as the quantization steps $L$ of the source ANN, the conversion error will be zero. An example is illustrated in Figure 2a, where $T = L = 4$, $\theta^l = \lambda^l$. The red curve presents the estimated output $\boldsymbol{\phi}^l(T)$ of the converted SNNs with respective to different input $\boldsymbol{z}^l$, while the green curve represents the out $\boldsymbol{a}^l$ of the source ANN with respective to different input $\boldsymbol{z}^l$. As the two curve are the same, the estimated conversion error $\widetilde{\boldsymbol{Err}}^l$ is zero. Nevertheless, in practical application, we focus on the performance of SNNs at different time-steps. There is no guarantee that the conversion error is zero when $T$ is not equal to $L$. As illustrated in Figure 2b, where $L = 4$ and $L = 8$, we can find the conversion error is greater than zero for some $\boldsymbol{z}^l$. This error will transmit layer-by-layer and eventually degrading the accuracy of the converted SNN. One way to solve this problem is to train multiple source ANNs with different quantization steps, then convert them to SNNs with different time-steps, but it comes at a considerable cost. In the next section, we propose the quantization clip-floor activation function with a shift term to solve this problem. Such an approach can achieve high accuracy for different time-steps, without extra computation cost.

### 4.2 QUANTIZATION CLIP-FLOOR-SHIFT ACTIVATION FUNCTION

We propose the quantization clip-floor-shift activation function to train ANNs.

$$\boldsymbol{a}^l = \hat{h}(\boldsymbol{z}^l) = \lambda^l \, \text{clip} \left( \frac{1}{L} \left\lfloor \frac{\boldsymbol{z}^l L}{\lambda^l} + \varphi \right\rfloor, 0, 1 \right). \tag{15}$$

Compared with Equation 13, there exists a hyperparameter vector $\varphi$ that controls the shift of the activation function. When $L \neq T$, we cannot guarantee the conversion error is 0. However, we can estimate the expectation of conversion error. Similar to (Deng & Gu, 2020), we assume that $z_i^l$ is uniformly distributed within intervals $[(t-1)\lambda^l/T, (t)\lambda^l/T]$ and $[(l-1)\lambda^l/L, (l)\lambda^l/L]$ for $t = 1, 2, ..., T$ and $L = 1, 2, ..., L$, we have the following Theorem.

**Theorem 2.** *An ANN with activation function (15) is converted to an SNN with the same weights. If $\theta^l = \lambda^l$, $\boldsymbol{v}^l(0) = \theta^l \varphi$, then for arbitrary $T$ and $L$, the expectation of conversion error reaches $\boldsymbol{0}$ when the shift term $\varphi$ in source ANN is $\frac{1}{2}$.*

$$\forall\, T, L \quad \mathbb{E}_z \left( \widetilde{\boldsymbol{Err}}^l \right)\Big|_{\varphi = \frac{1}{2}} = \boldsymbol{0}. \tag{16}$$

The proof is in the Appendix. Theorem 2 indicates that the shift term $\frac{1}{2}$ is able to optimize the expectation of conversion error. By comparing Figure 2b and Figure 2c, we can find that when the shift term $\varphi = 0.5$ is added, the mean conversion error reaches zero, even though $L \neq T$. These results indicate we can achieve high-performance converted SNN at ultra-low time-steps.

$L$ is the only undetermined hyperparameter of the quantization clip-floor-shift activation. When $T = L$, the conversion error reaches zero. So we naturally think that the parameter $L$ should be set as small as possible to get better performance at low time-steps. However, a too low quantization of the activation function will decrease the model capacity and further lead to accuracy loss when the time-steps is relatively large. Choosing the proper $L$ is a trade-off between the accuracy at low latency and the best accuracy of SNNs. We will further analyze the effects of quantization steps $L$ in the experiment section.

### 4.3 ALGORITHM FOR TRAINING QUANTIZATION CLIP-FLOOR-SHIFT ACTIVATION FUNCTION

Training an ANN with quantization clip-floor-shift activation instead of ReLU is also a tough problem. To direct train the ANN, we use the straight-through estimator (Bengio et al., 2013) for the derivative of the floor function, that is $\frac{\mathrm{d}\lfloor x \rfloor}{\mathrm{d}x} = 1$. The overall derivation rule is given in Equation 17.

$$\frac{\partial \widehat{h}_i(\boldsymbol{z}^l)}{\partial z_i^l} = \begin{cases} 1, & \text{if } -\frac{\lambda^l}{2L} < z_i^l < \lambda^l - \frac{\lambda^l}{2L} \\ 0, & \text{otherwise} \end{cases} \quad , \frac{\partial \widehat{h}_i(\boldsymbol{z}^l)}{\partial \lambda^l} = \begin{cases} \frac{1}{2L}, & \text{if } -\frac{\lambda^l}{2L} < z_i^l < \lambda^l - \frac{\lambda^l}{2L} \\ -\frac{z_i^l}{(\lambda^l)^2}, & \text{otherwise} \end{cases} \tag{17}$$

Here $z_i^l$ is the i-th element of $\boldsymbol{z}^l$. Then we can train the ANN with quantization clip-floor-shift activation using Stochastic Gradient Descent algorithm (Bottou, 2012).

## 5 RELATED WORK

The study of ANN-SNN conversion is first launched by Cao et al. (2015). Then Diehl et al. (2015) converted a three-layer CNN to an SNN using data-based and model-based normalization. To obtain high-performance SNNs for complex datasets and deeper networks, Rueckauer et al. (2016) and Sengupta et al. (2019) proposed more accurate scaling methods to normalize weights and scale thresholds respectively, which were later proved to be equivalent (Ding et al., 2021). Nevertheless, the converted deep SNN requires hundreds of time steps to get accurate results due to the conversion error analyzed in Sec. 3. To address the potential information loss, Rueckauer et al. (2016) and Han et al. (2020) suggested using "reset-by-subtraction" neurons rather than "reset-to-zero" neurons. Recently, many methods have been proposed to eliminate the conversion error. Rueckauer et al. (2016) recommended 99.9% percentile of activations as scale factors, and Ho & Chang (2020) added the trainable clipping layer. Besides, Han et al. (2020) rescaled the SNN thresholds to avoid the improper activation of spiking neurons. Massa et al. (2020) and Singh et al. (2021) evaluated the performance of converted SNNs on the Loihi Neuromorphic Processor. Our work share similarity with Deng & Gu (2020); Li et al. (2021), which also shed light on the conversion error. Deng & Gu (2020) minimized the layer-wise error by introducing extra bias in addition to the converted SNN biases. Li et al. (2021) further proposed calibration for weights and biases using quantized fine-tuning. They got good results with 16 and 32 time-steps without trails for more extreme time-steps. In comparison, our work aims to fit ANN into SNN with techniques eliminating

(a) VGG-16 on CIFAR-10    (b) ResNet-20 on CIFAR-10    (c) VGG-16 on CIFAR-100    (d) ResNet-20 on CIFAR-100

Figure 3: Compare ANNs accuracy.

the mentioned conversion error. The end-to-end training of quantization layers is implemented to get better overall performance. Our shift correction can lead to a single SNN which performs well at both ultra-low and large time-steps. Maintaining SNN performance within extremely few time-steps is difficult even for supervised learning methods like backpropagation through time (BPTT). BPTT usually requires fewer time-steps because of thorough training, yet at the cost of heavy GPU computation (Wu et al., 2018; 2019; Lee et al., 2016; Neftci et al., 2019; Lee et al., 2020; Zenke & Vogels, 2021). The timing-based backpropagation methods (Bohte et al., 2002; Tavanaei et al., 2019; Kim et al., 2020) could train SNNs over a very short temporal window, e.g. over 5-10 time-steps. However, they are usually limited to simple datasets like MNIST (Kheradpiseh & Masquelier, 2020) and CIFAR10 (Zhang & Li, 2020). Rathi et al. (2019) shortened simulation steps by initializing SNN with conversion method and then tuning SNN with STDP. In this paper, the proposed method achieves high-performance SNNs with ultra-low latency (4 time-steps).

## 6 EXPERIMENTS

In this section, we validate the effectiveness of our method and compare our method with other state-of-the-art approaches for image classification tasks on CIFAR-10 (LeCun et al., 1998), CIFAR-100 (Krizhevsky et al., 2009), and ImageNet datasets (Deng et al., 2009). Similar to previous works, we utilize VGG-16 (Simonyan & Zisserman, 2014), ResNet-18 (He et al., 2016), and ResNet-20 network structures for source ANNs. We compare our method with the state-of-the-art ANN-SNN conversion methods, including Hybrid-Conversion (HC) from Rathi et al. (2019), RMP from Han et al. (2020), TSC from Han & Roy (2020), RNL from Ding et al. (2021), ReLUThresholdShift (RTS) from Deng & Gu (2020), and SNN Conversion with Advanced Pipeline (SNNC-AP) from Li et al. (2021). Comparison with different SNN training methods is also included to manifest the superiority of low latency inference, including HybridConversion-STDB (HC-STDB) from Rathi et al. (2019), STBP from Wu et al. (2018), DirectTraining (DT) from Wu et al. (2019), and TSSL from Zhang & Li (2020). The details of the proposed ANN-SNN algorithm and training configurations are provided in the Appendix.

### 6.1 TEST ACCURACY OF ANN WITH QUANTIZATION CLIP-FLOOR-SHIFT ACTIVATION

We first compare the performance of ANNs with quantization clip-floor activation (green curve), ANNs with quantization clip-floor-shift activation (blue curve), and original ANNs with ReLU activation (black dotted line). Figure 3(a)-(d) report the results about VGG-16 on CIFAR-10, ResNet-20 on CIFAR-10, VGG-16 on CIFAR-100 and ResNet-20 on CIFAR-100. The performance of ANNs with quantization clip-floor-shift activation is better than ANNs with quantization clip-floor activation. These two ANNs can achieve the same performance as original ANNs with ReLU activation when $L > 4$. These results demonstrate that our quantization clip-floor-shift activation function hardly affects the performance of ANN.

### 6.2 COMPARISON WITH THE STATE-OF-THE-ART

Table 2 compares our method with the state-of-the-art ANN-SNN conversion methods on CIFAR-10. As for low latency inference ($T \leq 64$), our model outperforms all the other methods with the same time-step setting. For $T = 32$, the accuracy of our method is slightly better than that of ANN (95.54% vs. 95.52%), whereas RMP, RTS, RNL, and SNNC-AP methods have accuracy loss of 33.3%, 19.48%, 7.42%, and 2.01%. Moreover, we achieve an accuracy of 93.96% using only 4 time-steps, which is 8 times faster than SNNC-AP that takes 32 time-steps. For ResNet-20, we achieve an accuracy of 83.75% with 4 time-steps. Notably, our ultra-low latency performance is comparable with other state-of-the-art supervised training methods, which is shown in Table S3 of the Appendix.

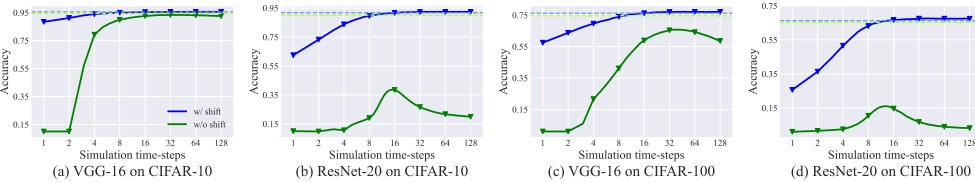

Figure 4: Compare quantization clip-floor activation with/without shift term

Table 2: Comparison between the proposed method and previous works on CIFAR-10 dataset.

| Architecture | Method | ANN | T=2 | T=4 | T=8 | T=16 | T=32 | T=64 | T≥512 |
|---|---|---|---|---|---|---|---|---|---|
| VGG-16 | RMP | 93.63% | - | - | - | - | 60.30% | 90.35% | 93.63% |
| | TSC | 93.63% | - | - | - | - | - | 92.79% | 93.63% |
| | RTS | 95.72% | - | - | - | - | 76.24% | 90.64% | 95.73% |
| | RNL | 92.82% | - | - | - | 57.90% | 85.40% | 91.15% | 92.95% |
| | SNNC-AP | 95.72% | - | - | - | - | 93.71% | 95.14% | 95.79% |
| | **Ours** | 95.52% | 91.18% | 93.96% | 94.95% | 95.40% | 95.54% | 95.55% | 95.59% |
| ResNet-20 | RMP | 91.47% | - | - | - | - | - | - | 91.36% |
| | TSC | 91.47% | - | - | - | - | - | 69.38% | 91.42% |
| | **Ours** | 91.77% | 73.20% | 83.75% | 89.55% | 91.62% | 92.24% | 92.35% | 92.41% |
| ResNet-18 | RTS [1] | 95.46% | - | - | - | - | 84.06% | 92.48% | 94.42% |
| | SNNC-AP [1] | 95.46% | - | - | - | - | 94.78% | 95.30% | 95.45% |
| | **Ours** | 96.04% | 75.44% | 90.43% | 94.82% | 95.92% | 96.08% | 96.06% | 96.06% |

[1] RTS and SNNC-AP use altered ResNet-18, while ours use standard ResNet-18.

We further test the performance of our method on the large-scale dataset. Table 3 reports the results on ImageNet, our method also outperforms the others both in terms of high accuracy and ultra-low latency. For ResNet-34, the accuracy of the proposed method is 4.83% higher than SNNC-AP and 69.28% higher than RTS when $T = 32$. When the time-steps is 16, we can still achieve an accuracy of 59.35%. For VGG-16, the accuracy of the proposed method is 4.83% higher than SNNC-AP and 68.356% higher than RTS when $T = 32$. When the time-steps is 16, we can still achieve an accuracy of 50.97%. These results demonstrate that our method outperforms the previous conversion methods. More experimental results on CIFAR-100 is in Table S4 of the Appendix.

### 6.3 COMPARISON OF QUANTIZATION CLIP-FLOOR AND QUANTIZATION CLIP-FLOOR-SHIFT

Here we further compare the performance of SNNs converted from ANNs with quantization clip-floor activation and ANN with quantization clip-floor-shift activation. In Sec. 4, we prove that the expectation of the conversion error reaches 0 with quantization clip-floor-shift activation, no matter whether $T$ and $L$ are the same or not. To verify these, we set $L$ to 4 and train ANNs with quantization clip-floor activation and quantization clip-floor-shift activation, respectively. Figure 4 shows how the accuracy of converted SNNs changes with respect to the time-steps $T$. The accuracy of the converted SNN (green curve) from ANN with quantization clip-floor activation (green dotted line) first increases and then decreases rapidly with the increase of time-steps, because we cannot guarantee that the conversion error is zero when $T$ is not equal to $L$. The best performance is still lower than source ANN (green dotted line). In contrast, the accuracy of the converted SNN from ANN with quantization clip-floor-shift activation (blue curve) increases with the increase of $T$. It gets the same accuracy as source ANN (blue dotted line) when the time-steps is larger than 16.

### 6.4 EFFECT OF QUANTIZATION STEPS L

In our method, the quantization steps $L$ is a hyperparameter, which affects the accuracy of the converted SNN. To analyze the effect of $L$ and better determine the optimal value, we train VGG-16/ResNet-20 networks with quantization clip-floor-shift activation using different quantization steps L, including 2,4,8,16 and 32, and then converted them to SNNs. The experimental results on CIFAR-10/100 dataset are shown in Table S2 and Figure 5, where the black dotted line denotes the ANN accuracy and the colored curves represent the accuracy of the converted SNN. In order to

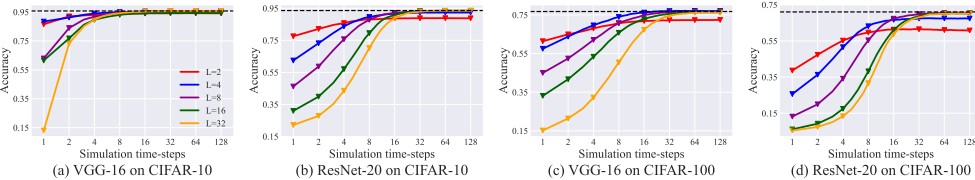

Figure 5: Influence of different quantization steps

Table 3: Comparison between the proposed method and previous works on ImageNet dataset.

| Architecture | Method | ANN | T=16 | T=32 | T=64 | T=128 | T=256 | T≥1024 |
|---|---|---|---|---|---|---|---|---|
| ResNet-34 | RMP | 70.64% | - | - | - | - | - | 65.47% |
| | TSC | 70.64% | - | - | - | - | 61.48% | 65.10% |
| | RTS | 75.66% | - | 0.09% | 0.12% | 3.19% | 47.11% | 75.08% |
| | SNNC-AP | 75.66% | - | 64.54% | 71.12% | 73.45% | 74.61% | 75.45% |
| | **Ours** | 74.32% | 59.35% | 69.37% | 72.35% | 73.15% | 73.37% | 73.39% |
| VGG-16 | RMP | 73.49% | - | - | - | - | 48.32% | 73.09% |
| | TSC | 73.49% | - | - | - | - | 69.71% | 73.46% |
| | RTS | 75.36% | - | 0.114% | 0.118% | 0.122% | 1.81% | 73.88% |
| | SNNC-AP | 75.36% | - | 63.64% | 70.69% | 73.32% | 74.23% | 75.32% |
| | **Ours** | 74.29% | 50.97% | 68.47% | 72.85% | 73.97% | 74.22% | 74.32% |

balance the trade-off between low latency and high accuracy, we evaluate the performance of converted SNN mainly in two aspects. First, we focus on the SNN accuracy at ultra-low latency (within 4 time-steps). Second, we consider the best accuracy of SNN. It is obvious to find that the SNN accuracy at ultra-low latency decreases as $L$ increases. However, a too small $L$ will decrease the model capacity and further lead to accuracy loss. When $L = 2$, there exists a clear gap between the best accuracy of SNN and source ANN. The best accuracy of SNN approaches source ANN when $L > 4$. In conclusion, the setting of parameter $L$ mainly depends on the aims for low latency or best accuracy. The recommend quantization step $L$ is 4 or 8, which leads to high-performance converted SNN at both small time-steps and very large time-steps.

## 7 DISCUSSION AND CONCLUSION

In this paper, we present ANN-SNN conversion method, enabling high-accuracy and ultra-low-latency deep SNNs. We propose the quantization clip-floor-shift activation to replace ReLU activation, which hardly affects the performance of ANNs and is closer to SNNs activation. Furthermore, we prove that the expected conversion error is zero, no matter whether the time-steps of SNNs and the quantization steps of ANNs is the same or not. We achieve state-of-the-art accuracy with fewer time-steps on CIFAR-10, CIFAR-100, and ImageNet datasets. Our results can benefit the implementations on neuromorphic hardware and pave the way for the large-scale application of SNNs.

Different from the work of Deng & Gu (2020), which adds the bias of the converted SNNs to shift the theoretical ANN-SNN curve to minimize the quantization error, we add the shift term in the quantization clip-floor activation function, and use this quantization clip-floor-shift function to train the source ANN. We show that the shift term can overcome the performance degradation problem when the time-steps and the quantization steps are not matched. Due to the unevenness error, there still exists a gap between ANN accuracy and SNN accuracy, even when $L = T$. Moreover, it is hard to achieve high-performance ANN-SNN conversion when the time-steps $T = 1$. All these problems deserve further research. One advantage of conversion-based methods is that they can reduce the overall computing cost while maintaining comparable performance as source ANN. Combining the conversion-based methods and model compression may help significantly reduce the neuron activity and thus reduce energy consumptions without suffering from accuracy loss (Kundu et al., 2021; Rathi & Roy, 2021), which is a promising direction.

ACKNOWLEDGEMENT

This work was supported by the National Natural Science Foundation of China under contracts No.62176003 and No.62088102.

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

# A APPENDIX

## A.1 NETWORK STRUCTURE AND TRAINING CONFIGURATIONS

Before training ANNs, we first replace max-pooling with average-pooling and then replace the ReLU activation with the proposed quantization clip-floor-shift activation (Equation 15). After training, we copy all weights from the source ANN to the converted SNN, and set the threshold $\theta^l$ in each layer of the converted SNN equal to the maximum activation value $\lambda^l$ of the source ANN in the same layer. Besides, we set the initial membrane potential $\boldsymbol{v}^l(0)$ in converted SNN as $\boldsymbol{\theta^l}/2$ to match the optimal shift $\boldsymbol{\varphi} = \frac{1}{2}$ of quantization clip-floor-shift activation in the source ANN.

Despite the common data normalization, we use some data pre-processing techniques. For CIFAR datasets, we resize the images into $32 \times 32$, and for ImageNet dataset, we resize the image into $224 \times 224$. Besides, we use random crop images, Cutout (DeVries & Taylor, 2017) and AutoAugment (Cubuk et al., 2019) for all datasets.

We use the Stochastic Gradient Descent optimizer (Bottou, 2012) with a momentum parameter of 0.9. The initial learning rate is set to 0.1 for CIFAR-10 and ImageNet, and 0.02 for CIFAR-100. A cosine decay scheduler (Loshchilov & Hutter, 2016) is used to adjust the learning rate. We apply a $5 \times 10^{-4}$ weight decay for CIFAR datasets while applying a $1 \times 10^{-4}$ weight decay for ImageNet. We train all models for 300 epochs. The quantization steps $L$ is set to 4 when training all the networks on CIFAR-10, and VGG-16, ResNet-18 on CIFAR-100 dataset. When training ResNet-20 on CIFAR-100, the parameter $L$ is set to 8. When training ResNet-34 and VGG-16 on ImageNet, the parameter $L$ is set to 8, 16, respectively. We use constant input when evaluating the converted SNNs.

## A.2 INTRODUCTION OF DATASETS

**CIFAR-10.** The CIFAR-10 dataset (Krizhevsky et al., 2009) consists of 60000 $32 \times 32$ images in 10 classes. There are 50000 training images and 10000 test images.

**CIFAR-100.** The CIFAR-100 dataset (Krizhevsky et al., 2009) consists of 60000 $32 \times 32$ images in 100 classes. There are 50000 training images and 10000 test images.

**ImageNet.** We use the ILSVRC 2012 dataset (Russakovsky et al., 2015), which consists 1,281,167 training images and 50000 testing images.

## A.3 DERIVATION OF EQUATION 12 AND PROOF OF THEOREM 2

**Derivation of Equation 11**

Similar to $\boldsymbol{z}^l = \boldsymbol{W}^l \boldsymbol{a}^{l-1}$ , We define

$$\boldsymbol{u}^l(t) = \boldsymbol{W}^l \boldsymbol{x}^{l-1}(t). \tag{S1}$$

We use $u_i^l(t)$ and $z_i^l$ to denote the $i$-th element in vector $\boldsymbol{u}^l(t)$ and $\boldsymbol{z}^l$, respectively. To derive Equation 11, some extra assumptions on the relationship between ANN activation value and SNN postsynaptic potentials are needed, which are showed in Equation S2.

$$\begin{cases} \text{if } z_i^l < 0, & \text{then } \forall t \; u_i^l(t) < 0, \\ \text{if } 0 \leqslant z_i^l \leqslant \theta_l, & \text{then } \forall t \; 0 \leqslant u_i^l(t) \leqslant \theta_l, \\ \text{if } z_i^l > \theta_l, & \text{then } \forall t \; u_i^l(t) > \theta_l. \end{cases} \tag{S2}$$

With the assumption above, we can discuss the firing behavior of the neurons in each time-step. When $z_i^l < 0$ or $z_i^l > \theta_l$, the neuron will never fire or fire all the time-steps, which means $\phi_i^l(T) = 0$ or $\phi_i^l(T) = \theta^l$. In this situation, we can use a clip function to denote $\phi_i^l(T)$.

$$\phi_i^l(T) = \text{clip}(z_i^l, 0, \theta^l). \tag{S3}$$

When $0 < z_i^l < \theta_l$, every input from the presynaptic neuron in SNNs falls into $[0, \theta^l]$, then we have $\forall t, \ v_i^l(t) \in [0, \theta]$. We can rewrite Equation 8 into the following equation.

$$\frac{\phi_i^l(T)T}{\theta^l} = \frac{z_i^l T + v_i^l(0)}{\theta^l} - \frac{v_i^l(T)}{\theta^l}. \tag{S4}$$

Considering that $\frac{\phi_i^l(T)T}{\theta^l} = \sum_{t=1}^{T} s_i^l(t) \in \mathbb{N}$ and $0 < \frac{v_i^l(T)}{\theta^l} < 1$, Equation S4 is changed to:

$$\phi_i^l(T) = \frac{\theta^l}{T} \left\lfloor \frac{z_i^l T + v_i^l(0)}{\theta^l} \right\rfloor. \tag{S5}$$

We combine these two situations (Equation S3 and Equation S4), and we have:

$$\boldsymbol{\phi}^l(T) = \theta^l \ \text{clip} \left( \frac{1}{T} \left\lfloor \frac{\boldsymbol{z}^l T + \boldsymbol{v}^l(0)}{\theta^l} \right\rfloor, 0, 1 \right). \tag{S6}$$

**Proof of Theorem 2**

Before prove Theorem 2, we first introduce Lemma 1.

**Lemma 1.** *If random variable $x \in [0, \theta]$ is uniformly distributed in every small interval $[m_t, m_{t+1}]$ with the probability density function $p_t$ ($t = 0, 1, ..., T$), where $m_0 = 0, m_{T+1} = \theta, m_t = \frac{(t-\frac{1}{2})\theta}{T}$ for $t = 1, 2, ..., T$, $p_0 = p_T$, we can conclude that*

$$\mathbb{E}_x \left( x - \frac{\theta}{T} \left\lfloor \frac{Tx}{\theta} + \frac{1}{2} \right\rfloor \right) = 0. \tag{S7}$$

*Proof.*

$$
\begin{aligned}
\mathbb{E}_x \left( x - \frac{\theta}{T} \left\lfloor \frac{Tx}{\theta} + \frac{1}{2} \right\rfloor \right) &= \int_0^{\theta/2T} p_0 \left( x - \frac{\theta}{T} \left\lfloor \frac{xT}{\theta} + \frac{1}{2} \right\rfloor \right) \ \mathrm{d}x \\
&+ \sum_{t=1}^{T-1} \int_{(2t-1)\theta/2T}^{(2t+1)\theta/2T} p_t \left( x - \frac{\theta}{T} \left\lfloor \frac{xT}{\theta} + \frac{1}{2} \right\rfloor \right) \ \mathrm{d}x \\
&+ \int_{(2T-1)\theta/2T}^{\theta} p_T \left( x - \frac{\theta}{T} \left\lfloor \frac{xT}{\theta} + \frac{1}{2} \right\rfloor \right) \ \mathrm{d}x \\
&= p_0 \int_0^{\theta/2T} x \ \mathrm{d}x + \sum_{t=1}^{T-1} p_t \int_{(2t-1)\theta/2T}^{(2t+1)\theta/2T} (x - \frac{t\theta}{T}) \ \mathrm{d}x + p_T \int_{(2T-1)\theta/2T}^{\theta} (x - \theta) \ \mathrm{d}x \\
&= p_0 \frac{\theta^2}{8T^2} + 0 - p_T \frac{\theta^2}{8T^2} = (p_0 - p_T) \frac{\theta^2}{8T^2} = 0. \tag{S8}
\end{aligned}
$$

$\square$

**Theorem 2.** *An ANN with activation function (15) is converted to an SNN with the same weights. If $\theta^l = \lambda^l$, $\boldsymbol{v}^l(0) = \theta^l \varphi$, then for arbitrary $T$ and $L$, the expectation of conversion error reaches $\boldsymbol{0}$ when the shift term $\varphi$ in source ANN is $\frac{1}{2}$.*

$$\forall \ T, L \quad \mathbb{E}_z \left( \widetilde{\boldsymbol{Err}}^l \right) \Big|_{\varphi = \frac{1}{2}} = \boldsymbol{0}. \tag{S9}$$

*Proof.*

$$\mathbb{E}_z \left( \widetilde{\boldsymbol{Err}}^l \right) \Big|_{\varphi = \frac{1}{2}} = \mathbb{E}_z \left( \frac{\theta^l}{T} \left\lfloor \frac{\boldsymbol{z}^l T + \boldsymbol{v}^l(0)}{\theta^l} \right\rfloor - \frac{\lambda^l}{L} \left\lfloor \frac{\boldsymbol{z}^l L}{\lambda} + \varphi \right\rfloor \right). \tag{S10}$$

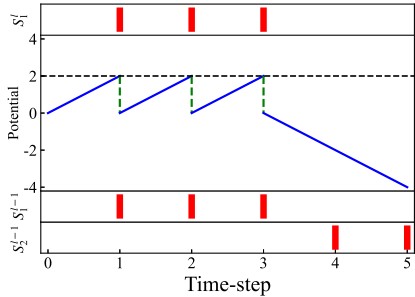

Figure S1: More spikes than expected exists for the method of setting the maximum activation.

As every element in vector $\boldsymbol{z}$ is identical, we only need to consider one element.

$$
\mathbb{E}_{z_i}\left(\frac{\theta^l}{T}\left\lfloor\frac{z_i^l T + v_i^l(0)}{\theta^l}\right\rfloor - \frac{\lambda^l}{L}\left\lfloor\frac{z_i^l L}{\lambda} + \varphi_i\right\rfloor\right)
$$
$$
= \mathbb{E}_{z_i}\left(\frac{\theta^l}{T}\left\lfloor\frac{z_i^l T + v_i^l(0)}{\theta^l}\right\rfloor - z_i^l\right) + \mathbb{E}_{z_i}\left(z_i^l - \frac{\lambda^l}{L}\left\lfloor\frac{z_i^l L}{\lambda} + \varphi_i\right\rfloor\right). \tag{S11}
$$

According to Lemma 1, we have

$$
\mathbb{E}_{z_i}\left(\frac{\theta^l}{T}\left\lfloor\frac{z_i^l T + v_i^l(0)}{\theta^l}\right\rfloor - z_i^l\right)\Bigg|_{v_i^l(0)=1/2} = 0, \tag{S12}
$$

$$
\mathbb{E}_{z_i}\left(z_i^l - \frac{\lambda^l}{L}\left\lfloor\frac{z_i^l L}{\lambda} + \varphi_i\right\rfloor\right)\Bigg|_{\varphi=1/2} = 0. \tag{S13}
$$

Thus the sum of both terms also equals zero. □

## A.4 COMPARISON OF THE METHODS WITH OR WITHOUT DYNAMIC THRESHOLD ON THE CIFAR-100 DATASET

In this paper we use a training parameter $\lambda^l$ to decide the maximum value of ANN activation. The previous works suggested to set the maximum value of ANN activation after training as the threshold. If we set $\theta^l = \max_{\boldsymbol{s}\in\{0,1\}^n}\left(\max(\theta^{l-1}\boldsymbol{W}^l\boldsymbol{s})\right)$, the situation of fewer spikes as expected never happens, as we can prove that $\boldsymbol{v}^l(T) < \theta^l$ (see Theorem 3). Despite this, there still exists the situation of more spikes as expected. An example is given in Figure S1. Here we consider the same example as in Figure 1. In source ANN, we suppose that two analog neurons in layer $l-1$ are connected to an analog neuron in layer $l$ with weights 2 and -2, and the output vector $\boldsymbol{a}^{l-1}$ of neurons in layer $l-1$ is $[0.6, 0.4]$. Besides, in converted SNN, we suppose that the two spiking neurons in layer $l-1$ fire 3 spikes and 2 spikes in 5 time-steps (T=5), respectively, and the threshold $\theta^{l-1} = 1$. Thus, $\phi^{l-1}(T) = \frac{\sum_{i=1}^T \boldsymbol{s}^{l-1}(i)}{T}\theta^{l-1} = [0.6, 0.4]$. According to Equation 1, the ANN output $\boldsymbol{a}^l = \boldsymbol{W}^l\boldsymbol{a}^{l-1} = [2, -2][0.6, 0.4]^T = 0.4$. As for SNN, we suppose that the presynaptic neurons fires at $t = 1, 2, 3$ and $t = 4, 5$, respectively. Even through we set the threshold $\theta^l = 1$ to the maximum activation 2, the postsynaptic neuron will fire three spikes at $t = 1, 2, 3$, and $\phi^l(T) = 0.6 > \boldsymbol{a}^l$.

Besides, setting $\max_{\boldsymbol{s}\in\{0,1\}^n}\left(\max(\theta^{l-1}\boldsymbol{W}^l\boldsymbol{s})\right)$ as the threshold brings two other problems. First, the spiking neurons will take a long time to fire spikes because of the large value of the threshold, which makes it hard to maintain SNN performance within a few time-steps. Second, the quantization error will be large as it is proportional to the threshold. If the conversion error is not zero for one layer, it will propagate layer by layer and will be magnified by larger quantization errors. We compare our method and the method of setting the maximum activation on the CIFAR-100 dataset. The results are reported in Table S1, where DT represents the dynamic threshold in our method. The results show that our method can achieve better performance.

Table S1: Comparison between our method and the method of setting the maximum activation.

| DT[1] | w/o shift | T=4 | T=8 | T=16 | T=32 | T=64 | T=128 | T=256 | T≥512 |
|---|---|---|---|---|---|---|---|---|---|
| **VGG-16 on CIFAR-100 with L=4** | | | | | | | | | |
| ✓ | ✓ | 69.62% | 73.96% | 76.24% | 77.01% | 77.10% | 77.05% | 77.08% | 77.08% |
| ✓ | ✗ | 21.57% | 41.13% | 58.92% | 65.38% | 64.19% | 58.60% | 52.99% | 49.41% |
| ✗ | ✓ | 1.00% | 0.96% | 1.00% | 1.10% | 2.41% | 13.76% | 51.70% | 77.10% |
| ✗ | ✗ | 1.00% | 1.00% | 0.90% | 1.00% | 1.01% | 2.01% | 19.59% | 70.86% |

[1] Dynamic threshold.

**Theorem 3.** *If the threshold is set to the maximum value of ANN activation, that is $\theta^l = \max_{\boldsymbol{s} \in \{0,1\}^n} \left( \max(\theta^{l-1} \boldsymbol{W}^l \boldsymbol{s}) \right)$, and $v_i^l(0) < \theta^l$. Then at any time-step, the membrane potential of each neuron after spike $v_i^l(t)$ will be less than $\theta^l$, where $i$ represents the index of each neuron.*

*Proof.* We prove it by induction. For $t = 0$, it is easy to see $v_i^l(0) < \theta^l$. For $t > 0$, we suppose that $v_i^l(t - 1) < \theta^l$. Since we have set the threshold to the maximum possible input, and $x_i^{l-1}(t)$ represents the input from layer $l - 1$ to the $i$-th neuron in layer $l$, $x_i^{l-1}(t)$ will be no larger than $\theta^l$ for arbitrary $t$. Thus we have

$$m_i^l(t) = v_i^l(t-1) + x_i^{l-1}(t) < \theta^l + \theta^l = 2\theta^l, \tag{S14}$$

$$s_i^l(t) = H(m_i^l(t) - \theta^l), \tag{S15}$$

$$v_i^l(t) = m_i^l(t) - s_i^l(t)\theta^l. \tag{S16}$$

If $\theta^l \leqslant m_i^l(t) < 2\theta^l$, then we have $v_i^l(t) = m_i^l(t) - \theta^l < \theta^l$. If $m_i^l(t) < \theta_l$, then $v_i^l(t) = m_i^l(t) < \theta_l$. By mathematical induction, $v_i^l(t) < \theta^l$ holds for any $t \geqslant 0$. □

## A.5 Effect of quantization steps L

Table S2 reports the performance of converted SNNs with different quantization steps $L$ and different time-steps $T$. For VGG-16 and quantization steps $L = 2$, we achieve an accuracy of 86.53% on CIFAR-10 dataset and an accuracy of 61.41% on CIFAR-100 dataset with 1 time-steps. When the quantization steps $L = 1$, we cannot train the source ANN.

## A.6 Comparison with state-of-the-art supervised training methods on CIFAR-10 dataset

Notably, our ultra-low latency performance is comparable with other state-of-the-art supervised training methods. Table S3 reports the results of hybrid training and backpropagation methods on CIFAR-10. The backpropagation methods require sufficient time-steps to convey discriminate information. Thus, the list methods need at least 5 time-steps to achieve ∼91% accuracy. On the contrary, our method can achieve 94.73% accuracy with 4 time-steps. Besides, the hybrid training method requires 200 time-steps to obtain 92.02% accuracy because of further training with STDB, whereas our method achieves 93.96% accuracy with 4 time-steps.

## A.7 Comparison on CIFAR-100 dataset

Table S4 reports the results on CIFAR-100, our method also outperforms the others both in terms of high accuracy and ultra-low latency. For VGG-16, the accuracy of the proposed method is 3.46% higher than SNNC-AP and 69.37% higher than RTS when $T = 32$. When the time-steps is only 4, we can still achieve an accuracy of 69.62%. These results demonstrate that our method outperforms the previous conversion methods.

Table S2: Influence of different quantization steps.

| quantization steps | T=1 | T=2 | T=4 | T=8 | T=16 | T=32 | T=64 | T=128 |
|---|---|---|---|---|---|---|---|---|
| **VGG-16 on CIFAR-10** | | | | | | | | |
| L=2 | 86.53% | 91.98% | 93.00% | 93.95% | 94.18% | 94.22% | 94.18% | 94.14% |
| L=4 | 88.41% | 91.18% | 93.96% | 94.95% | 95.40% | 95.54% | 95.55% | 95.59% |
| L=8 | 62.89% | 83.93% | 91.77% | 94.45% | 95.22% | 95.56% | 95.74% | 95.79% |
| L=16 | 61.48% | 76.76% | 89.61% | 93.03% | 93.95% | 94.24% | 94.25% | 94.22% |
| L=32 | 13.05% | 73.33% | 89.67% | 94.13% | 95.31% | 95.66% | 95.73% | 95.77% |
| **ResNet-20 on CIFAR-10** | | | | | | | | |
| L=2 | 77.54% | 82.12% | 85.77% | 88.04% | 88.64% | 88.79% | 88.85% | 88.76% |
| L=4 | 62.43% | 73.2% | 83.75% | 89.55% | 91.62% | 92.24% | 92.35% | 92.35% |
| L=8 | 46.19% | 58.67% | 75.70% | 87.79% | 92.14% | 93.04% | 93.34% | 93.24% |
| L=16 | 30.96% | 39.87% | 57.04% | 79.5% | 90.87% | 93.25% | 93.44% | 93.48% |
| L=32 | 22.15% | 27.83% | 43.56% | 70.15% | 88.81% | 92.97% | 93.48% | 93.48% |
| **VGG-16 on CIFAR-100** | | | | | | | | |
| L=2 | 61.41% | 64.96% | 68.0% | 70.72% | 71.87% | 72.28% | 72.35% | 72.4% |
| L=4 | 57.5% | 63.79% | 69.62% | 73.96% | 76.24% | 77.01% | 77.1% | 77.05% |
| L=8 | 44.98% | 52.46% | 62.09% | 70.71% | 74.83% | 76.41% | 76.73% | 76.73% |
| L=16 | 33.12% | 41.71% | 53.38% | 65.76% | 72.80% | 75.6% | 76.37% | 76.36% |
| L=32 | 15.18% | 21.41% | 32.21% | 50.46% | 67.32% | 74.6% | 76.18% | 76.24% |
| **ResNet-20 on CIFAR-100** | | | | | | | | |
| L=2 | 38.65% | 47.35% | 55.23% | 59.69% | 61.29% | 61.5% | 61.03% | 60.81% |
| L=4 | 25.62% | 36.33% | 51.55% | 63.14% | 66.70% | 67.47% | 67.47% | 67.41% |
| L=8 | 13.19% | 19.96% | 34.14% | 55.37% | 67.33% | 69.82% | 70.49% | 70.55% |
| L=16 | 6.09% | 9.25% | 17.48% | 38.22% | 60.92% | 68.70% | 70.15% | 70.20% |
| L=32 | 5.44% | 7.41% | 13.36% | 31.66% | 58.68% | 68.12% | 70.12% | 70.27% |

## A.8 ENERGY CONSUMPTION ANALYSIS

We evaluate the energy consumption of our method and the compared methods (Li et al., 2021; Deng & Gu, 2020) on CIFAR-100 datasets. Here we use the same network structure of VGG-16. Following the analysis in Merolla et al. (2014), we use synaptic operation (SOP) for SNN to represent the required basic operation numbers to classify one image. We utilize 77fJ/SOP for SNN and 12.5pJ/FLOP for ANN as the power consumption baseline, which is reported from the ROLLS neuromorphic processor (Qiao et al., 2015). Note that we do not consider the memory access energy in our study because it depends on the hardware. As shown in Table S5, when the time-steps is the same, the energy consumption of our method is about two times of SNNC-AP. However, to achieve the same accuracy of 73.55%, our method requires less energy consumption.

## A.9 PSEUDO-CODE FOR OVERALL CONVERSION ALGORITHM

In this section, we summarize the entire conversion process in Algorithm 1, including training ANNs from scratch and converting ANNs to SNNs. The QCFS in the pseudo-code represents the proposed quantization clip-floor-shift function.

Table S3: Compare with state-of-the-art supervised training methods on CIFAR-10 dataset

| Model | Method | Architecture | SNN Accuracy | Timesteps |
|---|---|---|---|---|
| **CIFAR-10** | | | | |
| HC | Hybrid | VGG-16 | 92.02 | 200 |
| STBP | Backprop | CIFARNet | 90.53 | 12 |
| DT | Backprop | CIFARNet | 90.98 | 8 |
| TSSL | Backprop | CIFARNet | 91.41 | 5 |
| DThIR[1] | ANN-SNN | cNet | 77.10 | 256 |
| **Ours** | ANN-SNN | VGG-16 | 93.96 | 4 |
| **Ours** | ANN-SNN | CIFARNet[2] | 94.73 | 4 |

[1] Implemented on Loihi neuromorphic processor
[2] For CIFARNet, we use the same architecture as Wu et al. (2018).

Table S4: Comparison between the proposed method and previous works on CIFAR-100 dataset.

| Architecture | Method | ANN | T=2 | T=4 | T=8 | T=16 | T=32 | T=64 | T≥512 |
|---|---|---|---|---|---|---|---|---|---|
| VGG-16 | RMP | 71.22% | - | - | - | - | - | - | 70.93% |
| | TSC | 71.22% | - | - | - | - | - | - | 70.97% |
| | RTS | 77.89% | - | - | - | - | 7.64% | 21.84% | 77.71% |
| | SNNC-AP | 77.89% | - | - | - | - | 73.55% | 76.64% | 77.87% |
| | **Ours** | 76.28% | 63.79% | 69.62% | 73.96% | 76.24% | 77.01% | 77.10% | 77.08% |
| ResNet-20 | RMP | 68.72% | - | - | - | - | 27.64% | 46.91% | 67.82% |
| | TSC | 68.72% | - | - | - | - | - | - | 68.18% |
| | **Ours** | 69.94% | 19.96% | 34.14% | 55.37% | 67.33% | 69.82% | 70.49% | 70.50% |
| ResNet-18 | RTS | 77.16% | - | - | - | - | 51.27% | 70.12% | 77.19% |
| | SNNC-AP | 77.16% | - | - | - | - | 76.32% | 77.29% | 77.25% |
| | **Ours** | 78.80% | 70.79% | 75.67% | 78.48% | 79.48% | 79.62% | 79.54% | 79.61% |

[1] RTS and SNNC-AP use altered ResNet-18, while ours use standard ResNet-18.

Table S5: Comparison of the energy consumption with previous works

| Method | | ANN | T=2 | T=4 | T=8 | T=16 | T=32 | T=64 |
|---|---|---|---|---|---|---|---|---|
| RTS | Accuracy | 77.89% | - | - | - | - | 7.64% | 21.84% |
| | OP (GFLOP/GSOP) | 0.628 | - | - | - | - | 0.508 | 0.681 |
| | Energy (mJ) | 7.85 | - | - | - | - | 0.039 | 0.052 |
| SNNC-AP | Accuracy | 77.89% | - | - | - | - | 73.55% | 76.64% |
| | OP (GFLOP/GSOP) | 0.628 | - | - | - | - | 0.857 | 1.22 |
| | Energy (mJ) | 7.85 | - | - | - | - | 0.660 | 0.094 |
| **Ours** | Accuracy | 76.28% | 63.79% | 69.62% | 73.96% | 76.24% | 77.01% | 77.10% |
| | OP (GFLOP/GSOP) | 0.628 | 0.094 | 0.185 | 0.364 | 0.724 | 1.444 | 2.884 |
| | Energy (mJ) | 7.85 | 0.007 | 0.014 | 0.028 | 0.056 | 0.111 | 0.222 |

---

**Algorithm 1** Algorithm for ANN-SNN conversion.

---

**Input**: ANN model $M_{\text{ANN}}(\boldsymbol{x}; \boldsymbol{W})$ with initial weight $\boldsymbol{W}$; Dataset $D$; Quantization step $L$; Initial dynamic thresholds $\boldsymbol{\lambda}$; Learning rate $\epsilon$.

**Output**: $M_{\text{SNN}}(\boldsymbol{x}; \hat{\boldsymbol{W}})$

1: **for** $l = 1$ to $M_{\text{ANN}}.$layers **do**
2:     **if** is ReLU activation **then**
3:         Replace ReLU$(\boldsymbol{x})$ by QCFS$(\boldsymbol{x}; L, \lambda^l)$
4:     **end if**
5:     **if** is MaxPooling layer **then**
6:         Replace MaxPooling layer by AvgPooling layer
7:     **end if**
8: **end for**
9: **for** $e = 1$ to epochs **do**
10:     **for** length of Dataset $D$ **do**
11:         Sample minibatch $(\boldsymbol{x}^0, \boldsymbol{y})$ from $D$
12:         **for** $l = 1$ to $M_{\text{ANN}}.$layers **do**
13:             $\boldsymbol{x}^l = $ QCFS$(\boldsymbol{W}^l \boldsymbol{x}^{l-1}; L, \lambda^l)$
14:         **end for**
15:         Loss $= $ CrossEntropy$(\boldsymbol{x}^l, \boldsymbol{y})$
16:         **for** $l = 1$ to $M_{\text{ANN}}.$layers **do**
17:             $\boldsymbol{W}^l \leftarrow \boldsymbol{W}^l - \epsilon \frac{\partial Loss}{\partial \boldsymbol{W}^l}$
18:             $\lambda^l \leftarrow \lambda^l - \epsilon \frac{\partial Loss}{\partial \lambda^l}$
19:         **end for**
20:     **end for**
21: **end for**
22: **for** $l = 1$ to $M_{\text{ANN}}.$layers **do**
23:     $M_{\text{SNN}}.\hat{\boldsymbol{W}}^l \leftarrow M_{\text{ANN}}.\boldsymbol{W}^l$
24:     $M_{\text{SNN}}.\theta^l \leftarrow M_{\text{ANN}}.\lambda^l$
25:     $M_{\text{SNN}}.\boldsymbol{v}^l(0) \leftarrow M_{\text{SNN}}.\theta^l / 2$
26: **end for**
27: **return** $M_{\text{SNN}}$

---

