# OpenReview forum: "Optimal ANN-SNN Conversion for High-accuracy and Ultra-low-latency Spiking Neural Networks"
_ICLR.cc/2022/Conference — ICLR 2022 Poster_

### Official Review · Reviewer_5DgR · 2021-10-30

**Correctness:** 4
**Technical Novelty And Significance:** 3
**Empirical Novelty And Significance:** 3
**Recommendation:** 8
**Confidence:** 5

**Main Review:**

Strengths:
(1) Excellently presented paper that provides a full explanation of the development in SNNs.
(2) The idea of obtaining a high-performance SNN from a quantization ANN is intuitive. However, the converted SNN will suffer a performance degradation problem if the time-steps of SNNs dose not match up with the quantization step of ANNs. The authors propose to add a shift term to overcome this problem, which is very interesting and impressive.
(3) The proposed framework is rigorous, and is supported by theoretical analysis.
(4) The proposed approach achieves SOTA accuracy with fewer time-steps.

Weaknesses & suggestions for improvement:
(1) My main concern is about unevenness error. Is there such a case, the input is negative during a period as the weight is negative? Thus, the spiking neurons fire fewer spikes than we expected.
(2) The accuracy of source ANNs on ImageNet is lower than some compared methods, which decreases the performance of converted SNNs.
(3) The accuracy of converted SNNs from ANNs with quantization clip-floor-shift activation (blue curve in Fig. 4c) is slightly higher than the source quantization ANN when the time-steps is large. I would like to know whether the performance is the same as or better than source ANN without quantization when the time-steps is large.
(4) I would like to see the discussion of the limitation of ANN-SNN conversion.


**Summary Of The Paper:**

This paper proposes a quantization clip-bottom-shift activation function to replace the ReLU activation function in ANNs, so as to better approximate the activation function of SNNs. The authors also prove that the expected error of ANN-SNN conversion can be reduced to 0 by using this method. The reported results on CIFAR-10/100 and ImageNet dataset show that this work archives state-of-the-art accuracy with fewer time-steps.

**Summary Of The Review:**

This paper proposes a novel ANN-SNN conversion framework and achieves SOTA accuracy with fewer time-steps. The suggestion for improvement is listed above.

---

> ### Author Response · Authors · 2021-11-19
> **Response to Reviewer 5DgR**
>
> Thank you for your positive and thoughtful comments. We are delighted that you find our idea very interesting and impressive, our framework rigorous, and the results state-of-the-art. We would like to address your concerns and answer your questions in the following.
>
> ### 1.  My main concern is about unevenness error. Is there such a case, the input is negative during a period as the weight is negative? Thus, the spiking neurons fire fewer spikes than we expected.
>
> We agree, the spiking neurons can fire fewer spikes than expected if the input is negative for a period and then positive. We have rewritten the section of unevenness error.  Unevenness error is caused by the unevenness of input spikes. If the timing of arrival spikes changes, the output firing rates will change, which causes conversion error. There are two situations: more spikes as expected or fewer spikes as expected. We have give an example in Figure 1 to illustrate it.
>
> ### 2.  The accuracy of source ANNs on ImageNet is lower than some compared methods, which decreases the performance of converted SNNs.
>
> Thanks for your comments. We have increased the accuracy of VGG-16 on ImageNet. The performance of converted SNNs also increases. Please refer to the section of To All Reviewers.
>
> ### 3. The accuracy of converted SNNs from ANNs with quantization clip-floor-shift activation (blue curve in Fig. 4c) is slightly higher than the source quantization ANN when the time-steps is large. I would like to know whether the performance is the same as or better than source ANN without quantization when the time-steps is large.
>
> The performance of the converted SNNs from ANNs with quantization clip-floor-shift activation is slightly lower than the source ANN without quantization. When the time-steps is large enough, the performance will be similar to the source ANN without quantization.
>
> ### 4. I would like to see the discussion of the limitation of ANN-SNN conversion.
>
> We have added the discussion of the limitation in the section of discussion and conclusion.

---

> > ### Comment · Reviewer_5DgR · 2021-11-27
> > **Response to authors**
> >
> > I would like to thank the authors for the clarifications. I am happy with the response. Overall, I think this paper is a nice submission. I would like to see the paper accepted.

---

### Official Review · Reviewer_hn22 · 2021-11-01

**Correctness:** 3
**Technical Novelty And Significance:** 3
**Empirical Novelty And Significance:** 3
**Recommendation:** 6
**Confidence:** 4

**Main Review:**

1.	There are other related works that need to be compared with this method. For example:

a.	R. Massa, A. Marchisio, M. Martina and M. Shafique, "An Efficient Spiking Neural Network for Recognizing Gestures with a DVS Camera on the Loihi Neuromorphic Processor," 2020 International Joint Conference on Neural Networks (IJCNN), 2020, pp. 1-9, doi: 10.1109/IJCNN48605.2020.9207109.

b.	S. Singh, A. Sarma, S. Lu, A. Sengupta, V. Narayanan and C. R. Das, "Gesture-SNN: Co-optimizing accuracy, latency and energy of SNNs for neuromorphic vision sensors," 2021 IEEE/ACM International Symposium on Low Power Electronics and Design (ISLPED), 2021, pp. 1-6, doi: 10.1109/ISLPED52811.2021.9502506.

2.	Figure 3 is not clear. It seems that the legend is missing to explain what is represented by the lines.
3.	Since they represent an important contribution, the Imagenet results should be reported and discussed in main manuscript, instead of placing them in the supplementary material.
4.	In Section 6, it can be observed that the ResNet-20 is the only considered CNN model in which the accuracy with T=4 is significantly lower than the accuracy with larger T. The reason should be discussed more comprehensively in the supporting text.


**Summary Of The Paper:**

This research proposes an ANN-SNN conversion method based on quantization clip-floor-shift activation.

**Summary Of The Review:**

The achieved experimental results are notable. However, some key points need to be addressed (see the main review section).

---

> ### Author Response · Authors · 2021-11-19
> **Response to Reviewer hn22**
>
> Thank you for your constructive comments. We are encouraged that you find our experimental results notable. We would like to address your concerns and answer your questions in the following.
>
> ### 1. There are other related works that need to be compared with this method.
>
> Thank you for your suggestion, we have added the comparison with [Massa et al., 2020] on the CIFAR-10 dataset. The results show that when the time-steps are the same, our method can achieve better performance. We also illustrate that the results of [Massa et al., 2020] are achieved on the Loihi Neuromorphic Processor. Besides, we have added the references of [Massa et al., 2020; Singh et al., 2021] in related work.
>
>
> ### 2. Figure 3 is not clear. It seems that the legend is missing to explain what is represented by the lines.
>
>  We have revised Figure 3. Here we want to illustrate that the ANNs with quantization clip-floor-shift activation can achieve the same performance as the original ANNs with ReLU activation when the quantization step $L>4$. The four sub-figures represent different ANN structures on different datasets, including VGG-16 on CIFAR-10 (a), ResNet-20 on CIFAR-10 (b), VGG-16 on CIFAR-100 \(c\) and ResNet-20 on CIFAR-100 (d). The green curve, blue curve, and black dotted line denote the performance of ANNs with quantization clip-floor activation, ANNs with quantization clip-floor-shift activation, and original ANNs with ReLU activation, respectively.
>
> ### 3. Since they represent an important contribution, the Imagenet results should be reported and discussed in main manuscript, instead of placing them in the supplementary material.
>
> We agree. We have moved the results on ImageNet to the main manuscript as you suggested. Please refer to the section of To All Reviewers.
>
> ### 4. In Section 6, it can be observed that the ResNet-20 is the only considered CNN model in which the accuracy with T=4 is significantly lower than the accuracy with larger T. The reason should be discussed more comprehensively in the supporting text.
>
> In our work, the ResNet-20 is trained with $L=8$ while the ResNet-18 and VGG-16 are trained with $L=4$. Therefore, the performance of the converted ResNet-20 is insignificant until $T\geqslant 8$. The reason to choose $L=8$ for ResNet-20 is that the model capacity of ResNet-20(\~0.27M Params) is much smaller than ResNet-18(\~11M Params) and VGG-16(\~34M Params). In fact, ResNet-20 is designed for CIFAR datasets while other architectures are designed for ImageNet dataset. Therefore, a relatively small quantization step $L$ will lead to poor ANN performance for ResNet-20. Details about the influence of quantization step $L$ on ANN accuracy is shown in Figure 3 in the manuscript and more discussions have been added to the revised paper.
>
> [Massa et al., 2020] Riccardo Massa, Alberto Marchisio, Maurizio Martina, and Muhammad Shafique. An efficient spiking neural network for recognizing gestures with a DVS camera on the Loihi neuromorphic processor. International Joint Conference on Neural Networks (IJCNN), 2020.
>
> [Singh et al., 2021] Sonali Singh, Anup Sarma, Sen Lu, Abhronil Sengupta, Vijaykrishnan Narayanan, and Chita R. Das. Gesture-SNN: co-optimizing accuracy, latency and energy of SNNs for neuromorphic vision sensors. In Proceedings of the ACM/IEEE International Symposium on Low Power Electronics and Design (ISLPED), 2021.

---

> > ### Comment · Reviewer_hn22 · 2021-11-29
> > **Response to authors**
> >
> > The effort made by the authors in addressing the reviewers' comments is appreciated. The paper can be accepted.

---

### Official Review · Reviewer_LAGa · 2021-11-01

**Correctness:** 3
**Technical Novelty And Significance:** 3
**Empirical Novelty And Significance:** 3
**Recommendation:** 6
**Confidence:** 4

**Main Review:**

Following are my detailed reviews.

Strenghts
========
1. The paper is well written and the general motivation of work on efficient ANN-SNN conversion is important as it can reduce the SNN training epoch over head. In particular, the general trend is that only ANN-SNN conversion asking for higher times steps (latency). So, achieving reduced latency for only conversion based approach is a promising direction.

2. The organization of the paper and results are good.

Weaknesses:
===========
1. The variable $\textbf{v}^l(t)$ is defined twice and are not having an exactly similar definition. Please clarify whether its membrane potential at time $t$ or membrane potential at $t$ after spike output happens. The definition of $\textbf{m}^l(t)$ is not clear either, what is meant by membrane potential after neuronal dynamics? I think, the latter is before applying the threshold ${\theta}^l$ and the earlier is after applying that as per Eq. 5. So, please clarify this part. Also, if you are not using the return to zero mechanisms, why is Eq. 5 valid?  Shouldn't it have ${\theta}^l_{reset}$? Or else clearly mention that $\theta^l$ is $\theta^l_{th}$ - ${\theta}^l_{reset}$.

2. An important way to reduce spiking activity is through weight sparsity. It would be interesting to see whether such "only conversion" based strategy go hand in hand with state-of-the-art model compression of SNNs [1, 2]. As, compression is a necessary part for model size reduction and is equally important as considering reduction of latency for edge deployment for real time applications.

3. I am not 100% sure about the way unevenness error has been portrayed here. I think its largely a function of what the max. weight value and the threshold is. In general we set the max. $W \times x$ value as the threshold. Where as in the example the authors has taken 1 as threshold and 1.5 as weight. Please also provide analysis with threshold as max ( $W \times x$). Also, I think the goal of trainable thresholds [2] should be to mitigate such error by default when SNN training happens. The authors should provide at least some discussion on that to clearly justify its inception and other ways of possible mitigation. Also, when the pre-synaptic spike is 1 at t=1, then the potential starts accumulating, then why it should immediately reflect at output at t = 1? If the inputs spikes at t = 1 then shouldn't the output postsynaptic spike happen at t = 2? I am not sure whether the Fig, 1(b) and (c) is correct. I think presynaptic red pulses should be at 0 and 2. If that's not the case, please justify why.

4. As the authors claimed the activation difference error is zero if the ANN is trained with the proposed activation non-linearity, then is it possible to convert with  T = 1? Please show results for T= 1 as that would make the limitations and possibilities much clearer.

5. Gu et al. [3] has also suggested ReLU shifting function in their optimal conversion work, how does the Therorem 2's shifting proposal differ from theirs, please highlight that.

6. One part may be confusing to the reader is that we generally evaluate threshold after ANN training, i that case how can we perform the proposed non-linear activation efficiently during ANN training. Please clearly state tis in the paper.

7. Please mention clearly how many training or test images you used for conversion.

[1]  Spike-thrift: Towards energy-efficient deep spiking neural networks by limiting spiking activity via attention-guided compression, WACV 2021.
[2] DIET-SNN: A LOW-LATENCY SPIKING NEURAL NETWORK WITH DIRECT INPUT ENCODING & LEAKAGE AND THRESHOLD OPTIMIZATION, IEEE TNNLS 2021.
[3] Optimal conversion of conventional artificial neural networks to spiking neural networks, ICLR 2021.


**Summary Of The Paper:**

The paper first analyzed the various sources of ANN-SNN conversion errors namely clipping error, quantization error and unevenness error and then proposed quantization clip-floor-shift activation function to replace the ReLU activation function in source ANNs to train them. This resulted in ANN-SNN conversion at ultra low latency (for example t step of 4).

**Summary Of The Review:**

The paper's effort to provide more similar ANN activation as SNN to reduce conversion error of ANN-SNN at low time step is interesting and promising. However, there are some concerns as I have detailed in my review that can further clarify the paper for good.

---

> ### Author Response · Authors · 2021-11-19
> **Response to Reviewer LAGa (Part 1/3)**
>
> Thank you for your insightful and very detailed feedback. We are delighted that you find our paper well-written and the results good. We would like to address your concerns and answer your questions in the following.
>
> ### 1. The variable $v^l(t)$ is defined twice and are not having an exactly similar definition. Please clarify whether its membrane potential at time $t$ or membrane potential at $t$ after spike output happens. The definition of $m^l(t)$ is not clear either, what is meant by membrane potential after neuronal dynamics? I think, the latter is before applying the threshold $\theta^l$ and the earlier is after applying that as per Eq. 5. So, please clarify this part. Also, if you are not using the return to zero mechanisms, why is Eq. 5 valid? Shouldn't it have $\theta^l_{rest}$? Or else clearly mention that $\theta^l$ is $\theta^l-\theta^l_{rest}$
>
> Thanks for pointing it. ${m}^l(t)$ and ${v}^{l}(t)$ represent the membrane potential before and after the trigger of a spike at time-step $t$. We have rewritten the section of Neuron model for SNNs to clarify it. To clariy all notations, we have added a table on page 3 to sort all notations out.
> In Equation (4), we used the same "reset-by-subtraction" mechanism as [Han et al., 2020; Deng and Gu, 2021], which means that after the neuron outputs a spike, the membrane potential will be reduced by an amount equal to the firing threshold $\theta^l$. Here we give an example to illustrate the difference between "reset-to-zero" mechanism and "reset-by-subtraction" mechanism. We suppose that the IF neurons in $l$-th layer receive the input ${x}^{l-1}(t)$ from last layer at time $t$, and the temporal potential ${m}_i^l(t)$ of the $i$-th neuron at time $t$ is large than the threshold $\theta^l$ (${m}_i^l(t)>\theta^l$). For "reset-to-zero" mechanism, the membrane potential after the trigger of a spike at time-step $t$ is zero, that is, ${v}_i^l(t)=0$. For "reset-by-subtraction" mechanism, the membrane potential after the trigger of a spike at time-step $t$ is large than zero, that is, ${v}_i^l(t)={m}_i^l(t)-\theta^l>0$.
>
> ### 2. An important way to reduce spiking activity is through weight sparsity. It would be interesting to see whether such "only conversion" based strategy go hand in hand with state-of-the-art model compression of SNNs.
>
> Thanks for your insightful suggestions. One advantage of conversion-based methods is that they can reduce the overall computing cost while maintaining comparable performance as source ANN. Combining the conversion-based methods and model compression may help significantly reduce the neuron activity and thus reduce energy consumptions without suffering from accuracy loss. We are glad to investigate it further in future work. We have added your suggestion in the section of discussion and conclusion.

---

> > ### Author Response · Authors · 2021-11-19
> > **Response to Reviewer LAGa (Part 2/3)**
> >
> > ### 3. I am not 100% sure about the way unevenness error has been portrayed here. I think its largely a function of what the max. weight value and the threshold is. In general we set the max. $W \times x$  value as the threshold. Where as in the example the authors has taken 1 as threshold and 1.5 as weight. Please also provide analysis with threshold as max ( $W \times x$ ). Also, I think the goal of trainable thresholds should be to mitigate such error by default when SNN training happens. The authors should provide at least some discussion on that to clearly justify its inception and other ways of possible mitigation.
> >
> > Thanks for your comments. We have rewritten the section of unevenness error. We are trying to show that the conversion error between ANNs and SNNs still exists, even though the clipping and quantization errors are zero, and we call this error as unevenness error. An example is shown in Figure 4 of the main text, where the green straight dotted line represents the accuracy of quantization clip-floor activation ANN with the quantization steps $L=4$, and the green curve represents the accuracy of converted SNNs. According to Equation (14), there is no clipping error here. The quantization error is also zero when the time-steps equals the quantization steps ($T=L=4$). However, one can find there is an obvious gap between ANN accuracy and SNN accuracy. We find this error is caused by the unevenness of input spikes. If the timing of arrival spikes changes, the output firing rates will change, which causes conversion error. There are two situations: more spikes as expected or fewer spikes as expected (shown in Figure 1 of the revised paper).
> >
> > We agree that if we set $\theta^l = max( W^l \times x )$, the situation of fewer spikes as expected never happens, as we can prove that $v^l(T)< \theta^l$. We have added the proof and more explanation in Appendix A.4. Despite this, there still exists the situation of more spikes as expected. An example is given in Figure S1 of Appendix A.4. Besides, setting max( $W \times x$ ) as the threshold brings two other problems. First, the spiking neurons will take a long time to fire spikes because of the large value of the threshold, which makes it hard to maintain SNN performance within a few time-steps. Second, the quantization error will be large as it is proportional to the threshold. If the conversion error is not zero for one layer, it will propagate layer by layer and will be magnified by larger quantization errors. Actually we have used a training parameter $\lambda^l$ to decide the maximum value of ANN activation. We have clarified the description and added the overall algorithm of ANN to SNN conversion in the revised paper. Besides, we have added the comparison between our method and the method of setting max( $W \times x$ ) as the threshold on the CIFAR-100 dataset in Tabel S1 of Appendix A.4. The results show that our method can achieve better performance.
> >
> > ### 4. When the pre-synaptic spike is 1 at t=1, then the potential starts accumulating, then why it should immediately reflect at output at t = 1? If the inputs spikes at t = 1 then shouldn't the output postsynaptic spike happen at t = 2? I am not sure whether the Fig, 1(b) and \(c\) is correct. I think presynaptic red pulses should be at 0 and 2. If that's not the case, please justify why.
> >
> > We agree that the output postsynaptic spike happens at $t=2$ if we consider the transmission delay. In this paper, we use the same neuron model in previous ANN-SNN works [Rathi et al., 2020; Han et al., 2020; Deng and Gu, 2021; Li et al., 2021], in which the transmission delay is ignored, that is, the postsynaptic neuron is designed to update the membrane potential immediately and output a spike as soon as the presynaptic neuron fires a spike. If we consider the transmission delay, it will take at least $l$ time-steps for an $l$-layer SNN to output spikes.
> >
> > ### 5. As the authors claimed the activation difference error is zero if the ANN is trained with the proposed activation non-linearity, then is it possible to convert with T = 1? Please show results for T=1 as that would make the limitations and possibilities much clearer.
> >
> > The results for $T=1$ on CIFAR-10 and CIFAR-100 datasets are shown in Table S2 of Appendix A.5 For VGG-16 and quantization steps $L=2$, we achieve an accuracy of 86.53\% on CIFAR-10 dataset and an accuracy of 61.41\% on CIFAR-100 dataset with 1 time-steps. When the quantization steps $L=1$, we cannot train the source ANN. We have added more discussion to clarify it.

---

> > > ### Author Response · Authors · 2021-11-19
> > > **Response to Reviewer LAGa (Part 3/3)**
> > >
> > > ### 6. Gu et al. has also suggested ReLU shifting function in their optimal conversion work, how does the Therorem 2's shifting proposal differ from theirs, please highlight that.
> > >
> > > Thanks for pointing it out. Please refer to the section of To All Reviewers.
> > >
> > > ### 7. One part may be confusing to the reader is that we generally evaluate threshold after ANN training, in that case how can we perform the proposed non-linear activation efficiently during ANN training. Please clearly state this in the paper.
> > >
> > > Our method modified the activation function of source ANN. As illustrated in Equation (13) and Equation (15) of the revised paper, we used a training parameter $\lambda^l$ to decide the maximum value of ANN activation. The output of ANN that is larger than $\lambda^l$ is clipped to $\lambda^l$. Therefore, we can train the ANNs with a modified activation function from scratch. Specifically, we train the weights and the parameter $\lambda^l$ of ANNs at the same time. When converting ANNs to SNNs, we set the threshold $\theta^l$ in each layer of the converted SNN equal to the maximum activation value $\lambda^l$ of the source ANN in the same layer. Thus, the threshold is determined during the ANN training process, not after. We have added the overall algorithm of ANN to SNN conversion in Appendix A.9 to clarify it.
> > >
> > > ### 8. Please mention clearly how many training or test images you used for conversion.
> > >
> > > Thanks for the suggestion. In Appendix A.2, we have added the description of the datasets we used. We train the source ANN from scratch and then convert the trained ANN to SNN. For the CIFAR-10 dataset, there are 50000 training images and 10000 test images. For the CIFAR-100 dataset, there are 50000 training images and 10000 test images. For the ImageNet dataset, there are 1,281,167 training images and 50000 test images.
> > >
> > >
> > > [Rathi et al., 2020] Nitin Rathi, Gopalakrishnan Srinivasan, Priyadarshini Panda, and Kaushik  Roy. Enabling Deep Spiking Neural Networks with Hybrid Conversion and Spike Timing Dependent Backpropagation. International Conference on Learning Representations (ICLR), 2020.
> > >
> > > [Han et al., 2020] Bing Han, Gopalakrishnan Srinivasan, and Kaushik Roy. RMP-SNN:Residual Membrane Potential Neuron for Enabling Deeper High-accuracy and Low-latency Spiking Neural Network. IEEE Conference on Computer Vision and Pattern Recognition (CVPR), 2020.
> > >
> > > [Deng and Gu, 2021] Shikuang Deng, and Shi Gu. Optimal Conversion of Conventional Artificial Neural Networks to Spiking Neural Metworks. International Conference on Learning Representations (ICLR), 2021.
> > >
> > > [Li et al., 2021] Yuhang Li, Shikuang Deng, Xin Dong, Ruihao Gong, and Shi Gu. A Free Lunch from Ann: Towards Efficient, Accurate Spiking Neural Networks Calibration. International Conference on Machine Learning (ICML), 2021.

---

> > > > ### Comment · Reviewer_LAGa · 2021-11-28
> > > > **Thanks for the rebuttal**
> > > >
> > > > Dear authors,
> > > >
> > > > Thanks for the rebuttal responses. My understanding from responses 7 and 8 is that you do not need a separate conversion as you can evaluate the threshold during ANN training. Thus no conversion training image set is necessary like [Rathi et al., 2020]. Also, by [Rathi et al., 2019] I hope you mean ICLR 2020, and not ICLR 2019! Please be watchful of the references you provide.
> > > >
> > > > I am overall convinced that this paper crosses the boundary of ICLR and thus I improve my score to 6.

---

> > > > > ### Author Response · Authors · 2021-11-29
> > > > > **Thanks for the Comment**
> > > > >
> > > > > Thanks for pointing it out! We have revised the references in the responses and will revise them in the paper.

---

### Official Review · Reviewer_d5di · 2021-11-02

**Correctness:** 2
**Technical Novelty And Significance:** 3
**Empirical Novelty And Significance:** 2
**Recommendation:** 6
**Confidence:** 5

**Main Review:**

**Pros**:

+ This paper gives a theoretical condition of zero conversion error. Under a uniform prior assumption, the authors propose to shift the activation and get zero expected error.

+ The empirical results show good improvements.

**Cons**:

- This is a little suggestion: this paper has too many notations, it would be better to have a section or table to sort them out.

- The analysis of conversion error has been proposed in Li et al. 2021. The only difference is the unevenness error where Li et al. 2021 assume it is 0. This paper is over-claiming and should point this out.

- What is the difference between shift term $\varphi$ in this paper and the optimal shift in Deng & Gu., 2020? Afaic, the uniform condition and the function of the shift are the same.

- Lack of ImageNet conversion experiments. The authors are strongly encouraged to convert more challenging models on the ImageNet dataset. Therefore, I would reject this paper for now.



**Summary Of The Paper:**

This paper proposes a Quantization neural network to spiking neural network (QNN2SNN) conversion method. The authors first analyze the conversion error between ANN and SNN. Then they construct the ann with quantized activation so that the error can be eliminated. Both theoretical and empirical results are presented in this paper.

**Summary Of The Review:**

My weakness is listed above. Besides, I also have some questions:

1. If Threorm 1 & 2 are correct, why your model does not reach exactly the same accuracy as ANN when $L=T$?
2. What is energy consumption? It is higher or lower?

---

> ### Author Response · Authors · 2021-11-19
> **Response to Reviewer d5di**
>
> Thank you for your thoughtful comments. We are encouraged you find that our experimental results show good improvements. We would like to address your concerns and your questions in the following.
>
> ### 1. This paper has too many notations, it would be better to have a section or table to sort them out.
>
> Thanks for your suggestion! We have added a table on page 3 to clarify all notations.
>
> ### 2. The analysis of conversion error has been proposed in Li et al. 2021. The only difference is the unevenness error where Li et al. 2021 assume it is 0. This paper is over-claiming and should point this out.
>
> Thanks for pointing it out! We agree that the clipping error and quantization error have been proposed in Li et al. 2021. The unevenness error is one contribution of our work. We find that the unevenness error, which is caused by the changes in the timing of arrival spikes and has been neglected in previous works, can induce more spikes or fewer spikes as expected. We have added the sentence "Note that the clipping error and quantization error have been proposed in Li et al. 2021" after introducing these conversions to emphasize this point. We also rewrite the summary of the first contribution in the introduction.
>
> ### 3. What is the difference between shift term $\varphi$ in this paper and the optimal shift in Deng & Gu., 2020? Afaic, the uniform condition and the function of the shift are the same.
>
> Thanks for your advice. We would like to clarify the difference. Please refer to the section of To All Reviewers. The uniform condition is inspired from Deng and Gu 2021, and we have mentioned that in our paper. We have added more discussion in the section of discussion and conclusion.
>
> ### 4. Lack of ImageNet conversion experiments. The authors are strongly encouraged to convert more challenging models on the ImageNet dataset.
>
> Thanks for the suggestion. We have compared our method with other state-of-the-art approaches on the ImageNet dataset. Please refer to the section of To All Reviewers.
>
> ### 5. If Threorm 1 & 2 are correct, why your model does not reach exactly the same accuracy as ANN when $L=T$?
>
> The gap between ANN accuracy and SNN accuracy is caused by unevenness error. When $L=T$, the quantization error and clipping error are zero, while the unevenness error still remains. The proposed shift term can decrease the unevenness error by increasing the initial potential of the converted SNN and alleviating the situation of fewer spikes as expected. We have added more discussion about it in the section of discussion and conclusion.
>
> ### 6. What is energy consumption? It is higher or lower?
>
> We have added analyses of energy consumption on the CIFAR-100 dataset. Here we use the same network structure of VGG-16. Following the analysis in [Merolla et al., 2014], we use synaptic operation (SOP) for SNN to represent the required basic operation numbers to classify one image. We utilize 77fJ/SOP for SNN as the power consumption baseline, which is reported from the ROLLS neuromorphic processor [Qiao et al., 2015]. Note that we do not consider the memory access energy in our study because it depends on the hardware. As shown in Table R2, when the time-steps is the same, the energy consumption of our method is about two times of SNNC-AP. However, to achieve the same accuracy of 73.55\%, our method requires less energy consumption. We have added the analyses of energy consumption in Appendix A.8.
>
>
> | Methods  | Items          | ANN   | T=2   | T=4   | T=8   | T=16  | T=32  | T=64  |
> | -------- | -------------- | ----- | ----- | ----- | ----- | ----- | ----- | ----- |
> | RTS      | Accuracy       | 77.89 | -     | -     | -     | -     | 7.64  | 21.84 |
> | SNNC-AP  | Accuracy       | 77.89 | -     | -     | -     | -     | 73.55 | 76.64 |
> | **Ours** | Accuracy       | 76.28 | 63.79 | 69.62 | 73.96 | 76.24 | 77.01 | 77.10 |
> | RTS      | OP (GFLOP/GSOP) | 0.314 | -     | -     | -     | -     | 0.508 | 0.681 |
> | SNNC-AP  | OP (GFLOP/GSOP) | 0.314 | -     | -     | -     | -     | 0.857 | 1.22  |
> | **Ours** | OP (GFLOP/GSOP) | 0.314 | 0.094 | 0.185 | 0.364 | 0.724 | 1.444 | 2.884 |
> | RTS      | Energy (mJ)    | 3.925 | -     | -     | -     | -     | 0.039 | 0.052 |
> | SNNC-AP  | Energy (mJ)    | 3.925 | -     | -     | -     | -     | 0.066 | 0.094 |
> | **Ours** | Energy (mJ)    | 3.925 | 0.007 | 0.014 | 0.028 | 0.056 | 0.111 | 0.222 |
> **Table R2: Comparison of power consumption.**
>
> [Merolla et al., 2014] Merolla, Paul A., et al. A million spiking-neuron integrated circuit with a scalable communication network and interface. Science. 2014, 34(6197): 668-673.
>
> [Qiao et al., 2015] Ning, Qiao, et al. A reconfigurable on-line learning spiking neuromorphic processor comprising 256 neurons and 128K synapses. Frontiers in neuroscience. 2015, 9: 141.

---

> > ### Comment · Reviewer_d5di · 2021-11-19
> > **A few more questions.**
> >
> > Thank you for your response and more experiments. I think your rebuttal did a good job in addressing my questions regarding empirical results. Therefore I increase my score to 5. However, I still have several theoretical questions:
> >
> > 1. I saw your notation table and it says $\boldsymbol{Err}^l$ is the conversion error, which, in my opinion, contains 3 types of error already. Regarding your $\boldsymbol{Err}^l$ in theorem I and 2, it is said to have *conversion error = 0* but your response claim *only quantization and clipping error is here*. Do I miss something?
> >
> > 2. Regarding your response to shift term. You mentioned that it is adopted in ANN training otherwise the performance is not good. However, in the recent trend of quantization, there is a type of works called Adaptive bitwidth [1], where they train the quantization with floor operation (no shift) and can adaptively adjust $L$. Is there any possibility that this shift-free scheme can be better, cause they can adapt different $L$
> >
> > [1] Jin, Q., Yang, L., & Liao, Z. (2020). Adabits: Neural network quantization with adaptive bit widths. In Proceedings of the IEEE/CVF Conference on Computer Vision and Pattern Recognition (pp. 2146-2156).

---

> > > ### Author Response · Authors · 2021-11-21
> > > **Response to Reviewer d5di's Further Comments**
> > >
> > > Thanks for your comments and suggestions.
> > >
> > > ### I saw your notation table and it says ${Err}^l$ is the conversion error, which, in my opinion, contains 3 types of error already. Regarding your $\pmb{Err}^l$in theorem I and 2, it is said to have conversion error = 0 but your response claim only quantization and clipping error is here. Do I miss something?
> > >
> > > Thanks for pointing out that the definition of $Err^l$ is confusing. We agree there are three types of conversion error in total. When estimating the output value of  converted SNNs in Equation (11), we assume that the potential $v^l(T)$ falls into $[0, \theta]$, so in this case, the unevenness error degenerates to the quantization error. Thus Equation (11) is an estimation of the conversion error. To clarify it, we have added $\widetilde{{Err}}^l$ to represent the estimated conversion error. The notation table is also updated. In fact, it is hard to estimate the unevenness error under general conditions. In this paper, the initial membrane potential in converted SNNs is set to half of the threshold， which can increase the firing rates of neurons and decrease the unevenness error by alleviating the situation of fewer spikes as expected.
> > >
> > > ### Regarding your response to shift term. You mentioned that it is adopted in ANN training otherwise the performance is not good. However, in the recent trend of quantization, there is a type of works called Adaptive bitwidth [Jin et al., 2020], where they train the quantization with floor operation (no shift) and can adaptively adjust. Is there any possibility that this shift-free scheme can be better, cause they can adapt different $L$.
> > >
> > > Thanks for your insightful suggestions. The adaptive bit-widths method [Jin et al., 2020] simultaneously trains models under different bit-widths with shared weights, which may further promote the accuracy of our framework when the time-steps $T$ and the quantization steps $L$ are not matched. However, there are also some issues to consider. First, the authors found that the 2-bit model does not converge for some cases, which brings challenges to implementing ultra-low-latency SNNs like $T=4$. Second, for each SNN, we are concerned about its performance for different time-steps $T$, ranging from 1 to 512. Considering the adaptive bit-widths method, we need to train many ANN models with different quantization steps $L$ and shared weights, which may not be an easy task. We are glad to investigate it further in future work.
> > >
> > > [Jin et al., 2020] Qing Jin, Linjie Yang, and Zhenyu Liao. Adabits: Neural Network Quantization with Adaptive Bit Widths. IEEE Conference on Computer Vision and Pattern Recognition (CVPR), 2020.

---

> > > > ### Comment · Reviewer_d5di · 2021-11-23
> > > > **Fair Enough**
> > > >
> > > > I want to thank the authors for their response. I think my concerns are addressed. I would increase my score to 6.

---

### Author Response · Authors · 2021-11-19
**To All Reviewers**

We sincerely thank all reviewers for insightful feedback. In this general response, we would like to address the concerns about performance on ImageNet and the comparison of shift terms.

### Performance on ImageNet

We have evaluated the performance of our method on ImageNet. In Table R1, we compare our method with RMP, TSC, RTS and SNNC-AP. One can find that our method outperforms the others both in terms of high accuracy and ultra-low latency. For ResNet-34, the accuracy of the proposed method is 4.83\% higher than SNNC-AP and 69.28\% higher than RTS when $T=32$. When the time-steps is 16, we can still achieve an accuracy of 59.35\%. For VGG-16, the accuracy of the proposed method is 4.83\% higher than SNNC-AP and 68.356\% higher than RTS when $T=32$. When the time-steps is 16, we can still achieve an accuracy of 50.97\%. These results demonstrate that our method outperforms the previous conversion methods.


| Arch. | Method   | ANN   | T=16  | T=32  | T=64  | T=128 | T=256 | T>=1024 |
| ------------ | -------- | ----- | ----- | ----- | ----- | ----- | ----- | ------- |
| ResNet-34    | RMP      | 70.64 | -     | -     | -     | -     | -     | 65.47   |
| ResNet-34    | TSC      | 70.64 | -     | -     | -     | -     | 61.48 | 65.10   |
| ResNet-34    | RTS      | 75.66 | -     | 0.09  | 0.12  | 3.19  | 47.11 | 75.08   |
| ResNet-34    | SNNC-AP  | 75.66 | -     | 64.54 | 71.12 | 73.45 | 74.61 | 75.45   |
| ResNet-34    | **Ours** | 74.32 | 59.35 | 69.37 | 72.35 | 73.15 | 73.37 | 73.39   |
| VGG-16       | RMP      | 73.49 | -     | -     | -     | -     | 48.32 | 73.09   |
| VGG-16       | TSC      | 73.49 | -     | -     | -     | -     | 69.71 | 73.46   |
| VGG-16       | RTS      | 75.36 | -     | 0.114 | 0.118 | 0.122 | 1.81  | 73.88   |
| VGG-16       | SNNC-AP  | 75.36 | -     | 63.64 | 70.69 | 73.32 | 74.23 | 75.32   |
| VGG-16       | **Ours** | 74.29 | 50.97 | 68.47 | 72.85 | 73.97 | 74.22 | 74.32   |

**Table R1: Comparison between the proposed method and previous works on ImageNet dataset.**


### Difference between the shift term ${\varphi}$ in this paper and the shift term in Deng and Gu, 2021.

We clarify the difference between the shift term of our work and that in [Deng and Gu, 2021] as follows:
The shift terms are added in different places. We add the shift term in the proposed quantization clip-floor activation function and use this quantization clip-floor-shift function to train the source ANN. The shift term not only plays a role in the inference process, but also influences the ANN training process. In the work of [Deng and Gu, 2021], they add $\theta^l/2T$ to the bias of the converted SNN, which helps to shift the theoretical ANN-SNN curve to minimize the quantization error (flooring error).

In our work, the initial membrane potential in converted SNNs is $\theta^l/2$, which corresponds to the shift term in source ANNs, whereas the initial membrane potential of SNNs in [Deng and Gu, 2021] is zero. There are also some similarities between these two approaches. For both methods, the aim of shift terms in SNNs is to make up for the influence of bias of SNNs caused by the floor function. As the shift term of converted SNNs at each time-steps in [Deng and Gu, 2021] is $\theta^l/2T$, the total shift of converted SNNs during the period from 0 to $T$ is $\theta^l/2$, which is the same as the initial membrane potential $\theta^l/2$ of our works. We believe that the setting of the initial membrane potential $\theta^l/2$ has unique advantages. Specifically, the firing rates of neurons can increase with the increase of the initial membrane potential, which can decrease the unevenness error by alleviating the situation of fewer spikes as expected. That may be one reason why our performance is better than [Deng and Gu, 2021].

Besides, the shift term in our work also has other functions. As we have shown in Figure 4, if we trained the quantization clip-floor ANNs without the shift term, the performance of converted SNNs will suffer a severe degradation problem when the time-steps $T$ and the quantization steps $L$ are not matched. By adding the shift term, the converted SNN can maintain high performance for nearly all time-steps $T$.
We have added more discussion about it in the section of discussion and conclusion.


[Deng and Gu, 2021] Shikuang Deng, and Shi Gu. Optimal Conversion of Conventional Artificial Neural Networks to Spiking Neural Metworks. International Conference on Learning Representations (ICLR), 2021.

---

### Decision · Program_Chairs · 2022-01-20

**Decision:**

Accept (Poster)

**Comment:**

The authors present an improved method to convert ANNs to spiking neural networks (SNNs). First, a network with quantized activations is constructed, then it is converted. They analyze the conversion errors theoretically. In addition to previously considered errors [Li et al. 2021] they also consider an error they call "unevenness error" and propose a way to compensate for that.
They test the method on data sets such as CIFAR-100 and show good improvements over previous methods with respect to classification accuracy and inference time.
The reviewers agree that the manuscript presents interesting and valuable work with a significant novel contribution.The manuscript is well written.

Weak points according to the first reviews were:
- Lack of ImageNet conversion experiments.
- Analysis of energy consumption was missing.
- More related work needs to be compared.
The revision addressed all these points, This was acknowledged by the reviewers with increased ratings. All reviewers propose acceptance.